# Order of magnitude wall time improvement of variational methane inversions by physical parallelization: a demonstration using TM5-4DVAR

Sudhanshu Pandey[1,*], Sander Houweling[2], Arjo Segers[3]

[1]SRON Netherlands Institute for Space Research, Utrecht, the Netherlands
[2]Department of Earth Sciences, Vrije Universiteit Amsterdam, Amsterdam, the Netherlands
[3]TNO Department of Climate, Air and Sustainability, Utrecht, The Netherlands

*now at Jet Propulsion Laboratory, California Institute of Technology, Pasadena, CA, USA

*Correspondence to*: Sudhanshu Pandey (s.pandey@sron.nl)

**Abstract.** Atmospheric inversions are used to constrain emissions of trace gases using atmospheric mole fraction measurements. The four-dimensional variational (4DVAR) inversion approach allows optimization of emissions at a higher temporal and spatial resolution than ensemble or analytical approaches but provides limited opportunities for scalable
parallelization because it is an iterative optimization method. Multidecadal variational inversions are needed to optimally extract information from the long measurement records of long-lived atmospheric trace gases like carbon dioxide and methane. However, the wall time needed—up to months—complicates these multidecadal inversions. The physical parallelization (PP) method introduced by Chevallier (2013) addresses this problem for carbon dioxide inversions by splitting the period of the chemical transport model into blocks and running them in parallel. Here we present a new implementation of the PP method
which is suitable for methane inversions accounting for the chemical sink of methane. The performance of the PP method is tested in an 11-year inversion using a TM5-4DVAR inversion setup that assimilates surface observations to optimize methane emissions at grid-scale. Our PP implementation improves the wall time performance by a factor of 5 and shows excellent agreement with a full serial inversion in an identical configuration (global mean emissions difference = 0.06 % with an interannual variation correlation R = 0.99; regional mean emission difference < 5 % and interannual variation R > 0.94). The
wall time improvement of the PP method increases with the size of the inversion period. The PP method is planned to be used in future releases of the Copernicus Atmosphere Monitoring Service (CAMS) multidecadal methane reanalysis.

## 1 Introduction

Methane ($CH_4$) is the second-most important greenhouse gas after carbon dioxide ($CO_2$), and its atmospheric abundance has increased by more than a factor of 2.5 since preindustrial times. Methane is responsible for 25 % of the anthropogenic radiative
forcing despite its 200 times lower abundance than $CO_2$ due to its strong global warming potential (Myhre et al., 2013). Atmospheric inversions provide methane emission estimates by optimally combining the information in atmospheric

observations and bottom-up emissions (estimates using process-based models and inventories) along with corresponding error characteristics. Chemical transport models (CTMs) simulate the spatiotemporal distribution of the methane mole fractions in the atmosphere for a given set of emissions while also accounting for its atmospheric sink. Inversions use CTMs to disentangle the influences of atmospheric transport from the influences of emissions and sinks on the observed mole fractions (Naus et al., 2019, Pandey et al., 2019). A few studies have performed inversions on multidecadal scales to constrain emissions using the long measurement record of methane mole fractions. For example, "The Global Methane Budget 2000–2017" (Saunois et al., 2020) presents regional emission estimates from nine different inversion setups. The methane emissions reanalysis project under the Copernicus Atmosphere Monitoring Service (CAMS) performs multidecadal inversions using the TM5-4DVAR variational approach to provide regularly updated gridded methane emissions (Segers and Houweling, 2020).

Trace gas inversions adjust a state vector, which includes gridded emissions (and sometimes initial mole fraction field and other parameters), to improve the agreement between model simulations and observations. The inversions minimize a Bayesian cost function that is defined based on the difference between the modelled and observed mole fractions as well as the magnitude of the emission adjustments, weighted with respective error covariances. There are three main approaches used in atmospheric inverse modelling: analytical, ensemble and variational. The analytical approach is based on a closed-form solution of Bayes' theorem (Gurney et al., 2002). It requires the calculation of observation-sensitivities of each of the state vector elements separately. The large computational cost involved restricts its application to small size state vectors. The ensemble approach improves the computational performance by parameterizing the state vector sensitivities using a statistical ensemble (Peters et al., 2005). Still only a relatively small size state vector can be afforded using this approach.

The variational inversion approach was introduced to lift the state vector size restriction (Chevallier et al., 2005). In this approach, the cost function minimum is computed using an iterative procedure, with each iteration comprising of a forward and an adjoint CTM run. The variational approach can be applied to weakly non-linear inverse problems using a suitable steepest-decent numerical minimizer (Naus et al., 2021; Pandey et al., 2016). Truncated posterior uncertainties can be obtained from variational inversions using the conjugate gradient minimizer for linear inverse problems (Meirink et al., 2008). A more robust, but computationally expensive, estimate of posterior uncertainties can be obtained using a Monte Carlo method (Chevallier et al., 2007, Pandey et al., 2016). However, as each iteration of a variational inversion uses the output of the previous iteration, the calculations can take months depending on the spatial and temporal resolution. This long wall time limits the resolution, duration, and number of iterations that can be used in multidecadal variational inversions. To reduce this long wall time for carbon dioxide ($CO_2$) inversions, Chevallier (2013) introduced the physical parallelization (PP) method. In this method, the full inversion period is split into a number of blocks, and the CTM runs for the blocks are performed in parallel within each iteration. Corrections are added to the simulated $CO_2$ mole fractions in a block to account for emission adjustments (iteration minus prior emissions) in earlier blocks. This method reduced the wall time by an order of magnitude (seven-fold improvement for a 32-year inversion) while keeping the inversion-derived emission adjustments statistically consistent with a

serial inversion. However, the original implementation of the PP method cannot be used for a reactive trace gas like methane as the method does not account for atmospheric chemical sink.

Here we present an improved PP method that accounts for the limited atmospheric lifetime of reactive trace gases such as
methane, which has an atmospheric lifetime of about 9 years (mainly due to oxidation by the OH radicals). The intention is to use this new PP implementation for the Copernicus Atmosphere Monitoring Service (CAMS) methane flux reanalysis, which aims to provide annually updated multidecadal emission estimates, within a production cycle of only a few months. In the next section, we present our PP method. The method's performance is tested using an 11-year inversion setup presented in Section 3. The wall time and optimized emissions of a PP inversion are compared to a serial inversion in an identical configuration. In
Section 4, we discuss possible future improvements and applications of the PP method. Our conclusions are summarized in Section 5.

## 2 Physical parallelization for methane inversions

An inversion of an atmospheric trace gas minimizes a Bayesian cost function of the state vector $x$:

$$J(x) = \frac{1}{2}[x - x^b]\mathbf{B}^{-1}[x - x^b] + \frac{1}{2}[m - y]\mathbf{R}^{-1}[m - y]. \quad \ldots\ldots(1).$$

Here $y$ is the observation vector, and $x^b$ is the prior state vector. $\mathbf{B}$ and $\mathbf{R}$ are the error covariance matrices of the prior emissions and the observations, respectively. The vector $m$ constitutes the modelled mole fractions corresponding to $y$. It is
computed using a CTM operator $H$, which simulates the mole fractions given the emissions in the state $x$ and the initial mole fraction field $c_0$ .

$$m = H(c_0, x) \ldots\ldots(2).$$

In a variational inversion setup, the posterior solution of Equation (1) is obtained by minimizing $J$ using an iterative procedure that computes a new emission update $x^{i+1}$ in each iteration $i$ using the gradient

$$\nabla J(x^i) = \mathbf{B}^{-1}[x^i - x^b] + H^*(\mathbf{R}^{-1}[m^i - y]) \ldots\ldots(3).$$

$H^*$ represents the adjoint CTM operator, which is implemented using the adjoint code of the CTM. The inversion finishes when a predefined convergence criterion is met, such as a desired gradient norm reduction or simply a maximum number of iterations.

In the PP method presented in Chevallier (2013), the full period of the inversion is split into $r$ overlapping time blocks, which
can be run in parallel. Figure 1 schematically represents the main steps in the PP method used in the forward mode to calculate $\boldsymbol{m}^i$. At the start of the inversion, a serial CTM run (without segmentation) is performed to calculate initial mole fraction fields $\boldsymbol{c}_k^b$ for each block $k$ using the prior emissions $\boldsymbol{x}^b$. In an iteration, the block mole fractions for the iteration $\boldsymbol{m}_k^i$ is computed using the block CTM operator $H_k$, the iteration emissions for the block $\boldsymbol{x}_k^i$, the initial mole fraction for this block $\boldsymbol{c}_k^b$, and a mole fraction correction $n_k^i$:


$$\boldsymbol{m}_k^i = H_k\left(\boldsymbol{c}_k^b, \boldsymbol{x}_k^i\right) + n_k^i \quad \ldots\ldots\ldots(4).$$

Here the scalar $n_k^i$ accounts for the global mean mole fraction changes due to emission differences $(\boldsymbol{x}^i - \boldsymbol{x}^b)$ during the inversion period that precedes the block $k$. The error due to this simplification is further reduced by using an overlap period
between consecutive blocks, where modelled mole fractions from the succeeding block are discarded. The overlap period CTM run distributes the emission differences uniformly through the Earth's atmosphere. The PP method by Chevallier (2013) was applied to $CO_2$ inversions, where the scalar mole fraction correction $n_k^i$ for block $k$ was simply calculated as the sum of the emission differences from each preceding blocks (i.e., block 1, 2, 3…..$k$-1):

$\quad n_k^i = \sum_{l=1}^{k-1} f\mathbf{E}\left[\boldsymbol{x}_l^i - \boldsymbol{x}_l^b\right] \quad \ldots\ldots (5)$

Here $\mathbf{E}$ denotes a summation matrix used to compute global sum of the elements of $\boldsymbol{x}_l$. $f$ is a scalar used to convert emissions to mole fractions assuming a uniform distribution of the emitted trace gas throughout the Earth's atmosphere. $f$ is calculated simply as the ratio between the number of moles in a unit emission and the number of moles of air in the atmosphere.

Methane has an atmospheric lifetime of about 9 years. Unlike $CO_2$, the mole fraction impact of methane emission differences will be reduced due to atmospheric chemistry within the duration of a multidecadal inversion as well as within a PP inversion block. Therefore, in our new implementation of the PP, we use a mole fraction correction vector $\boldsymbol{n}_k^i$ (with size of $\boldsymbol{m}_k^i$) instead of the scalar $n_k^i$ to apply separate corrections to each observation. We account for the limited lifetime of methane by
implementing an atmospheric sink operator $S$. In addition, we use a CTM block sensitivity vector $\boldsymbol{h}_k$ to distribute global emission changes more precisely, taking into account the full 3D atmospheric transport and the sink rather than assuming a

globally uniform distribution. $h_k$ is computed at the start of the inversion by running $H_k$ in forward mode with an uniform initial mole fraction field and zero emissions, i.e., $h_k = H_k (c_k = 1, x_k = 0)$. $n_k^i$ is computed as

$\quad n_k^i = h_k \sum_{l=1}^{k-1} s_{k,l} \; f \; \mathbf{E}\left[x_l^i - x_l^b\right]$ ...........(6).

Here the scalar $s_{k,l}$ accounts for the impact of atmospheric sinks on the global uniform mole fraction change during the time period between the blocks $k$ and $l$. $s_{k,l}$ is generated using a sink operator $S$. We describe a formulation of $S$ in the next section. Within block $k$ itself, the impact of atmospheric sinks is accounted for by $h_k$.


Each iteration of a variational inversion computes a departures vector $\delta m$ :

$\quad \delta m = \mathbf{R}^{-1}[m - y]$ .......... (7).

The adjoint CTM $H^*$ is run with $\delta m$ to compute the local gradient of the cost function (Equation 2). In the PP method, $H^*$ is split into blocks covering the same periods as used for the forward CTM simulation. In an iteration, each adjoint block is first run with the respective departures. Then, the modelled adjoint sensitivities of a block $\delta x_k^i$ are adjusted for the effects of departures of succeeding blocks by adding adjoint mole fraction correction scalar $g_k^i$:

$\quad g_k^i = f \sum_{l=k+1}^{r} s_{k,l} \; h_l^{\mathrm{T}} \; \delta m_l^i$ (8),

Here $h_l^{\mathrm{T}} \; \delta m_i^i$ is the matrix dot product of the two vectors, both of which have the same size. The correctness of the adjoint implementation of the PP method can be verified using the adjoint test (Meirink et al., 2008). The test checks if the equality

$\quad \langle M(a), b \rangle = \langle a, M^*(b) \rangle$ ....... (9),

is satisfied to an accuracy near the computing precision. $M$ and $M^*$ denote the forward and adjoint model operators, $\langle \; \rangle$ denotes the inner product. $a$ and $b$ are the arbitrary forward and adjoint model states.

In a PP inversion, the initial mole fraction field $c_0$ needs to be consistent with the observations as a discrepancy between the two leads to large emission differences in the early months of the inversion period. This issue can easily be dealt with in a serial inversion using a spin-up period and rejecting this period from the posterior solution. However, in a PP inversion, the large emission differences may result in large mole fraction corrections, which increases the error in the PP approximation

(see Equations 4 & 5). This can be avoided by taking a realistic $c_0$ from the posterior mole fractions simulations of another inversion covering the period before the PP inversion. If such an inversion is not available, $c_0$ can be computed by performing an inversion for the 1-year period preceding the PP inversion.

In summary, the main steps of the PP methane inversion are as follows:

1. Construct an initial mole fraction field $c_0$ consistent with observations at the start of the inversion.
2. Split the full period of the inversion into $r$ overlapping time blocks.
3. To calculate $c_k^b$, run the forward CTM serially with prior emissions $x^b$ and save the simulated mole fraction fields at the start time of each block.
4. Calculate the CTM block sensitivities vector $h_k$ by running the CTM over each block with a uniform initial mole fraction field of 1 and zero emissions, and sample the model output at the observation time and locations.
5. Prepare a sink operator $S$ which accounts for the impact of atmospheric sinks on methane mole fractions during a period.
6. Perform the inversion by iteratively minimizing the cost function until the convergence condition is met using a forward and an adjoint run in each iteration:
   a. Forward run:
      i. Run all forward CTM blocks in parallel with the initial mole fraction fields from step 3.
      ii. Account for the emission changes relative to the prior in preceding blocks by adding the corrections $n_k^i$ (Equation 5).
   b. Adjoint run:
      i. Run all adjoint CTM blocks in parallel to calculate the adjoint emission sensitivities.
      ii. Add the adjoint correction $g_k^i$ to account for the effect of departures in successive blocks (Equation 8).

The CTM runs in the steps 4, 6.a.i and 6.b.i are performed in parallel. The steps without CTM run (1, 2, 5, 6.a.ii and 6.b.ii) require very little wall time. Step 3 is the most time-consuming because a full serial CTM run is performed in the step.

**3 PP performance test**

We evaluate the performance of the PP method by comparing a PP inversion with a serial methane inversion. Both inversions are performed for an 11-year period (1999-2010) with identical observations and prior emissions. We use the TM5-4DVAR inversion system (Bergamaschi et al., 2010; Meirink et al., 2008, Krol et al., 2005) with the settings used in Pandey et al. (2016). The TM5 CTM is run at $6° \times 4°$ horizontal resolution and 25 vertical hybrid sigma-pressure levels from the surface to the top of the atmosphere. The meteorological fields for this offline model are taken from the European Centre for Medium-Range Weather Forecasts (ECMWF) ERA-Interim reanalysis (Dee et al., 2011). We optimize a single category ('total') of

methane emissions at 6° × 4° spatial resolution and monthly temporal resolution. The posterior emissions of the two inversions are compared after integrating over the TRANSCOM regions shown in Figure 2a.

The inversion assimilates surface observations from the NOAA Earth System Research Laboratory (ESRL) global cooperative air sampling network at on- and off-shore sites (Dlugokencky et al., 2011; Dlugokencky et al., 2020). The locations of the observation sites are shown in Figure 2b. The prior covariance matrix **B** is constructed as follows. The diagonal elements of **B** are constructed assuming ±1σ uncertainties of 50 % of the emissions per grid cell per month. The off-diagonal elements are constructed by assuming the emissions to be correlated temporally using an exponential correlation function with an e-folding time scale of 3 months, and spatially with a Gaussian correlation function using a length scale of 500 km (Houweling et al., 2014). Uncertainties of 1.4 ppb are assigned to methane observations. Our system also assigns a modelling representation error based on simulated local mole fractions gradients (Basu et al., 2013). The prior emissions are taken from the same sources as in Pandey et al. (2016). The prior emissions of the year 2008 are used for every year in the prior, hence there is no interannual variability in the prior emissions. The cost function $J$ is minimized using the conjugate gradient minimizer, which is based on the Lanczos algorithm (Fisher and Courtier, 1995). The inversions use the convergence criterion of gradient norm reduction by a factor 1000, which is achieved after 19 iterations in both inversions.

In the PP inversion, we split the inversion period of 1999-2010 into 11 blocks of 21 months. The first 9 months of each block is the overlap period used for uniformly mixing the emission changes within the atmosphere, while the last 12 months provide modelled mole fractions for assimilating the observations. We parameterize the sink operator $S$, which computes the sink scaling factor $s_{k,l}$ (Equation 6), with an e-folding decay function and a constant atmospheric lifetime of methane ($\tau$) of 9 years.

$$s_{k,l} = S(k,l) = e^{-|t_l - t_k|/\tau} \quad \dots\dots (10)$$

Here $t_l$ and $t_k$ are the start times of the blocks $l$ and $k$, respectively. We found this simple parameterization with a constant lifetime is sufficient for our test inversion.

The input emissions of TM5 are mass fluxes (Tg yr[-1]) and the output is in mole fractions (ppb). The methane emission changes are converted in mole fractions using an $f = 0.361 \frac{ppb}{Tg}$. Successful implementation of the PP method in the adjoint mode was verified using the adjoint test (Equation 9).

**3.1 PP inversion errors**

Here we evaluate the difference in modelled mole fractions and posterior emissions between the PP and serial inversions.

### 3.1.1 Mole fraction errors

We first examine the quality of the inversion-optimized fit to the observation. Figure 3 shows the time series of the prior and posterior simulations and the observations for two background sites, representing each hemisphere: Barrow (Alaska) and the South Pole. The prior root mean square difference (RMSD) with observations for Barrow (78 ppb) is 3 times higher than for the South Pole (28 pbb). Barrow observations show more high-frequency variations than the South Pole as the Northern Hemisphere station is influenced by methane emissions from wetlands. The mole fractions simulated by the PP inversion are in good agreement with the results obtained from the serial inversion: RMSDs of 2 ppb and 1 pbb for Barrow and the South pole, respectively, which are only 2.5% and 3.2% of the prior RMSD. This shows that the PP inversion, starting from an identical prior, is able to match the observations at these sites as well as the serial inversion.

Figure 4 shows the average mole fraction differences at all observation sites. The observation-prior RMSD for all observations combined is 67 ppb. The mean mismatch is –58 ppb because the 2008 bottom-up emissions used as the prior are larger than the mean posterior emission over 1999-2010. The average data uncertainty (mean of the square root of the diagonal elements of **R**) is 19 ppb (not shown). For both inversions, a good model fit (90 % reduction in mean of observation-model mole fraction mismatch) to the observations is achieved with a gradient norm reduction of 1000. The posterior simulation of both the serial and PP inversions reduce the RMSD to 20 ppb (mean = –2 ppb). The RMSD between PP and serial is 1.9 ppb (mean = –0.1 ppb), which is an order of magnitude smaller than the posterior-prior RMSD of 62 ppb (mean = –55 ppb). This shows that the implementation of the PP method has little impact on the inversion's ability to fit the observations.

### 3.1.2 Posterior emission errors

A good agreement between observations and posterior models does not guarantee that the inversions have produced similar posterior emissions. The physically parallelized CTM used in the PP inversion has lost some of the consistency of the full CTM used in the serial inversion and the PP emission errors will depend on the impact of this CTM simplification. Figure 5 shows mean emissions (averaged over 1999-2010) from the inversions integrated over the globe and TRANSCOM regions. We do not have a good estimate of the posterior uncertainties because a large number of variational inversion iterations are needed for the second derivative of the cost function to converge. Therefore, we evaluate PP performance by comparing the PP-serial emission differences against the emission adjustments performed by the serial inversion (serial-prior differences) and prior emission uncertainties. The serial inversion adjusts the global mean prior emissions of $544 \pm 11$ Tg yr$^{-1}$ by $-22$ Tg yr$^{-1}$. The PP inversion is in excellent agreement with the serial inversion in this respect. The two differ by 0.3 Tg yr$^{-1}$ (0.06%), which is 1% of the difference between the prior and serial emissions. The global methane emissions are in general well constrained by the NOAA observations in a serial inversion, and the additional error introduced by the PP method only causes a 1 % error relative to the serial-prior emission mismatch. At regional scales, the serial inversion adjustment is the smallest

for Australia: + 0.4 Tg yr$^{-1}$ from the prior emissions of 6.6 ± 0.4 Tg yr$^{-1}$. The PP inversion adjusts the prior emissions here by +0.5 Tg yr$^{-1}$, implying that the difference with the serial inversion (0.1 Tg yr$^{-1}$). The serial inversion changes the Eurasian temperate emissions the most, by –58 Tg yr$^{-1}$, where prior emissions are 135 ± 8 Tg yr$^{-1}$. The PP inversion changes these emissions by –60 Tg yr$^{-1}$, i. e., a difference of 2 Tg yr$^{-1}$. The South American temperate region has the largest PP error, relative to the serial-prior difference, of 2 Tg yr$^{-1}$ The serial emissions for this region are 6.5 Tg yr$^{-1}$ higher than the prior of 36 ± 2.4 Tg yr$^{-1}$. In summary, mean PP emission estimates for the TRANSCOM regions deviate within < 5 % from the prior emissions, while the serial-prior differences are up to 50 % of the prior emissions.

Figure 6 shows the inter-annual variability of the emission estimates. The global emissions time series of the PP and serial inversions show a very good agreement with a correlation coefficient R = 0.99, explained by the large observational constraint. Over the TRANSCOM regions, the North American temperate region has the best agreement (R = 1.0). All other regions have R higher than 0.98 except for Australia (0.96) and Europe (0.94). Figure 7 shows the intra-annual variations of the emissions. At the global scale, the PP and serial time series match very well with R = 1.00, whereas R between prior and serial is 0.93. The agreement between PP and serial time series is also very good for all TRANSCOM regions (R > 0.98) despite low correlations between prior and serial emissions for some regions, for example, R = 0.13 for the South American temperate region. This shows that the PP inversion is able to reproduce the seasonal cycle of the emissions very well. Figure 8 shows the spatial distribution of the emission differences at grid scale. The mean (± 1σ spread) of the differences between the serial inversion and prior is $-8 \times 10^{-3}$ (± 0.5 ) Tg gridbox$^{-1}$ yr$^{-1}$, and it is $9 \times 10^{-5}$ (± 0.04 ) Tg gridbox$^{-1}$ yr$^{-1}$ for serial and PP inversions. Emission differences between the PP and serial inversions are visible over India and South American temperate. These differences are likely due to the lack of observational constraint in these regions (see Figure 2). In summary, the combination of small differences in the mean emissions, and the high correlations between intra- and inter-annual time series, shows that the PP inversion can effectively reproduce results of the serial inversion at regional scales.

**3.2 Wall time**

Table 1 compares the wall times used by the PP and serial inversions. The TM5 model in our inversion setup uses OpenMP parallelization and gives the best wall time performance on 4 CPUs on a single node (12-core 2.6 GHz Intel Xeon E5-2690 v3). Using more CPUs reduces the performance as the communication overhead within the CPUs becomes the bottleneck (Note that the TM5-MP version described in Williams et al., 2017, with improved parallel scaling, was not used in this study). In this configuration, a forward or adjoint TM5 CTM run of one year took about 15 minutes. Hence an iteration of the serial inversion, consisting of 11 years forward and adjoint runs, required 5 hours. The PP inversion iterations were performed in 11 parallel blocks of 21 months each on 4 CPUs. A single PP iteration took 55 mins, which is > 5 times faster than the serial inversion. The main steps of PP implementation are listed in Section 2. In our inversion test, the initial mole fraction fields $c_0$ (step 1) were taken from an inversion using surface measurements that was not performed in this study. Steps 1, 2, 5, 6.a.ii

and 6.b.ii took negligible time. Step 3 took 2.5 hours because it consists of a full serial TM5 forward run. Steps 4, 6.a.i and 6.b.i consist of 11 TM5 run over blocks of 21 months which were run in parallel and took 25 minutes each. Note that an iteration took longer than the sum of the forward and adjoint block runs because of a few minutes waiting time for the computer cores to become available again. In total, the PP inversion took 20 hours, 5 times less than the serial inversion which took 101 hours. Note that although the PP inversion took a shorter wall time, it needed extra CPU core hours for the additional 9-month overlap, CTM block sensitivity and initial mole fraction computation runs. The PP inversion used a total of 700 CPU core hours, whereas the serial inversion used about 400 CPU core hours. Table 1 also provides a projection of the wall time improvement of a hypothetical 35-year inversion (not performed in this study) based on the TM5-4DVAR inversion setup used in this study. A PP inversion would be 15 times faster for such a long period. Overall, we find that the PP method, which accounts for the atmospheric lifetime of methane, is able to effectively reproduce the posterior emissions of a 11-year conventional serial inversion 5 times faster.

## 4 Discussion

### 4.1 PP method applications

The utility of the PP method for inversion of a trace gas depends on the time scale of the influence of emissions on observations within the spatial domain of the CTM. Therefore, PP is mainly useful in global inversions of trace gases that have atmospheric lifetime of a year or longer in the atmosphere. For a trace gas with a shorter lifetime, such as of carbon monoxide with 2 months lifetime, emission perturbations last for a short duration. A multidecadal inversion of such a trace gas can be broken into many short inversions. These short inversions can be performed in parallel, and the posterior emission can be combined thereafter. A similar approach can be used for regional inversions of short-and long-lived trace gases because emission perturbations are quickly advected out of the regional CTM domain and hence do not influence observations for a long period.

The hydroxyl radical OH is the main sink of methane in the atmosphere. Zhang et al. (2018) showed that the satellite-observed atmospheric signature of the methane sink is sufficiently distinct from that of methane emissions, hence OH mole fractions can be optimized using synthetic shortwave infrared (SWIR) and thermal infrared (TIR) satellite observations. Following up on this, Maasakkers et al. (2019) and Zhang et al. (2021) used methane observations from the GOSAT satellite to optimize atmospheric OH fields along with methane emissions. The simultaneous optimization of OH with methane emissions introduces a non-linearity in the inversion because methane loss rate depends on the product of methane and OH mole fractions. However, the changes to the methane mole fractions are expected to remain small during the inversions. Hence, the non-linear effect is small and a quasi-linearity is assumed to solve the inversion analytically using the computation of the full Jacobian matrix of the CTM. Under a quasi-linearity assumption, OH can be optimized in a PP methane inversion by introducing annual OH scaling factors in the state vector and the methane lifetimes in the sink operator can be scaled in each iteration to reflect

the corresponding OH adjustments. Such an implementation can also be used in inversions optimizing OH using methyl chloroform (Naus et al., 2021).


## 4.2 Possible further improvements

The PP method accounts for changes in the background mole fractions due to emission changes in preceding blocks using a sink operator $S$, a CTM block sensitivity $h$, and an overlap between the consecutive blocks. In our test experiment, $S$ is assumed to be an e-folding decay function with an atmospheric lifetime of methane of 9 years, which we found to be sufficient for the
annually-repeating OH field used in our 11-year CTM runs. Methane lifetime within the duration of a longer multidecadal inversion will vary due to climatological influences as well as possible trends and interannual variations in the hydroxyl radical abundance. In such cases, $S$ can be defined as a function of an annual lifetime vector for the specific CTM run. The lifetime vector can be calculated as the ratio of the annual sink and global methane burden simulated by the serial CTM run in step 3 of the PP method.

The overlap period between consecutive blocks in the PP method allows methane emission perturbations to mix within the CTM domain according to atmospheric transport. We used a 9-month overlap in our test inversion setup. It was sufficient estimate the total emissions from TRANSCOM regions using the surface observations. The 6-month overlap used by Chevalier et al. (2013) for $CO_2$ inversion was found to be insufficient for a PP methane inversion, likely because of the differences between the source and sink distributions of methane and $CO_2$. Increasing the overlap period to 9-month and using CTM block
sensitivity vector solved this issue. We expect that a 1-year overlap, equal to the interhemispheric mixing time, would be more than sufficient for all tracers irrespective of their source-sink distribution and lifetime. A shorter overlap would improve the computational efficiency and wall time but reduce the accuracy of the physical parallelization of the CTM. The PP accuracy could be maintained with shorter overlap periods by using a mole fraction correction vector per hemisphere rather than the single global vector used in this study. However, the computational resources and wall time saved by this would be partially
spent on the additional block sensitivity runs. Our test inversions are performed at a relatively coarse horizontal resolution of $6° \times 4°$ with 25 vertical hybrid sigma-pressure levels. We do not expect the performance of the PP method to degrade significantly for higher resolution inversions if there is sufficient overlap between the blocks and the mole fraction corrections are parameterized correctly. Furthermore, the performance gained by performing the inversions at higher resolution because of the improved computational performance will likely outweigh the accuracy loss due to the assumptions made in the PP
method.

The PP method reduced the wall time of the CTM simulations in a variational inversion but introduces additional model errors because of the simplifications made. For our test inversion setup, these PP-CTM model errors are minor as the posterior PP emission estimates are in good agreement with the serial estimates. In future PP implementations, these PP-CTM errors can

be accounted for in the observation error matrix **R**. The PP-CTM error can be calculated as the difference between the model
output of a PP and a serial forward CTM run with randomly perturbed prior emissions.

### 4.3 Current CAMS inversion setup

In the future, the PP method will be implemented in the CAMS multidecadal methane emissions reanalysis setup. The
European Commission has anticipated the need for reliable information about atmospheric composition of greenhouse gases
through development of numerical systems that combine sophisticated physical models with measurements from a wide range
of observing systems for an operational service, which is being implemented. The current CAMS methane flux reanalysis
product (Segers and Houweling, 2020) uses the TM5-4DVAR inverse modelling system and provides measurement-informed
monthly methane emission estimates. The latest release has two sets of methane emissions: (1) release v19r1 for 1990-2019
using surface observation; (2) release v19r1s for 2010-2019 using surface and also GOSAT satellite observations. The surface
observations are mainly from the NOAA network (Dlugokencky et al., 2011). Methane emissions are optimized at $3° \times 2°$
spatial resolution and monthly temporal resolution using TM5 with 34 vertical layers. If performed in serial mode each iteration
of the 1990-2019 inversion would take about 5-10 days, and the full inversion will require multiple months to finish. Segers
and Houweling (2020) circumvent this issue by breaking the full inversion into smaller inversions of 3-year time windows that
are performed in parallel. The target inversion on high resolution ($3° \times 2°$ degrees, 34 layers) is preceded by a coarse resolution
inversion ($6° \times 4°$, 25 layers) that provides the initial mole fraction fields and is processed serially. The high-resolution
inversion optimizes only the emissions and uses initial mole fractions for each 3-year block obtained from mole fraction fields
of a coarse resolution inversion, which optimizes both emission and initial mole fractions. The 1990-2019 inversion using this
approach still takes 3-4 months to finish and requires about 40 smaller inversions to provide the end result. These numbers
depend of course on the parallel efficiency of the model and the computing server. The need for a coarse resolution serial
sequence of inversions to provide initial mole fractions fields limits the inversion period for which this method can be used.
With the implementation of the PP method presented in this study, the wall time performance of the CAMS reanalysis
inversions will improve in future.

### 5 Conclusions

Regular surface observations of methane mole fractions started in early 1984, and by now the measurement record spans more
than 35 years (Dlugokencky et al., 2011). An atmospheric inversion with a very large state vector is needed optimize emissions
using such long measurement records at a grid scale. The variational inversion approach allows for optimization of a much
larger state vector than the ensemble or analytical approaches. However, each iteration of a variational inversion uses the

output of the previous iteration, limiting the opportunity for scalable parallelization. At the same time, an increase in the spatiotemporal resolution of CTMs needed to take full advantage of the rapidly improving precision and coverage of surface and satellite measurements results in a rapid increase in wall time.

We have developed the PP method for methane inversions which improves the wall time of variational methane inversions by physical CTM parallelization while accounting for the atmospheric lifetime in forward and adjoint variational modes. We have tested the performance of this method using an 11-year TM5-4DVAR inversion setup that consists of a conventional serial inversion and a PP inversion in an identical configuration. The PP method reduced the wall time by a factor of 5 while still showing excellent agreement with the posterior emissions from the serial inversion. The wall time improvement of using PP will be even larger for longer inversions, for example, by a factor of 15 for a 35-year inversion. The PP method makes multidecadal global inversions of long-lived atmospheric trace gases more feasible. It will be implemented in the CAMS reanalysis setup which provides regular updates of multidecadal emission estimates by assimilating surface and satellite observations.

*Data Availability*. NOAA ESRL methane observations used in this study are available on Zenodo in the input folder of the TM5-4DVAR-PP code (https://doi.org/10.5281/zenodo.6326373, Pandey et al., 2022).

*Code availability*. The TM5-4DVAR-PP version 1.0-beta-1 code used in this study for the simulations can be downloaded from Zenodo (https://doi.org/10.5281/zenodo.6326373, Pandey et al., 2022). The TM5 model is described in detail on http://tm5.sourceforge.net/.

*Author contributions*. The study was designed by SH and AJ. SP and AJ developed the PP. SP implemented the PP method on TM5-4DVAR and did the performance test simulations. SP wrote the manuscript using contributions from all the co-authors.

*Acknowledgements*. We thank the efforts of NOAA and other surface observations networks for producing and maintaining the vital long record of global methane observations. The computations for this study were carried out on the Dutch national supercomputer Cartesius (https://userinfo.surfsara.nl/systems/cartesius; last access: 27-03-2020) maintained by SURFSara. SP and AJ were funded for this study by the Copernicus Atmosphere Monitoring Service, implemented by the European Centre for Medium-Range Weather Forecasts on behalf of the European Commission (grant no. CAMS73).

*Competing interests*. The authors declare that they have no competing interests.

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

**Table 1** Wall time comparison for the inversions performed in this study. Wall time projections for a hypothetical 35-year inversion are also given.

| Model runs | | Serial | PP |
|---|---|---|---|
| One year forward or adjoint run | | 15 minutes | |
| 1999-2010 inversion | 1 iteration (forward + adjoint TM5 run) | 5 hours | 55 minutes |
| | Inversion with 19 iterations | 101 hours | 20 hours |
| 1985-2020 inversion* | 1 iteration (forward + adjoint TM5 run) | 16 hours | 55 minutes |
| | Inversion with 50 iterations | 34 days | 56 hours |

*Projection based on the 1999-2010 inversion


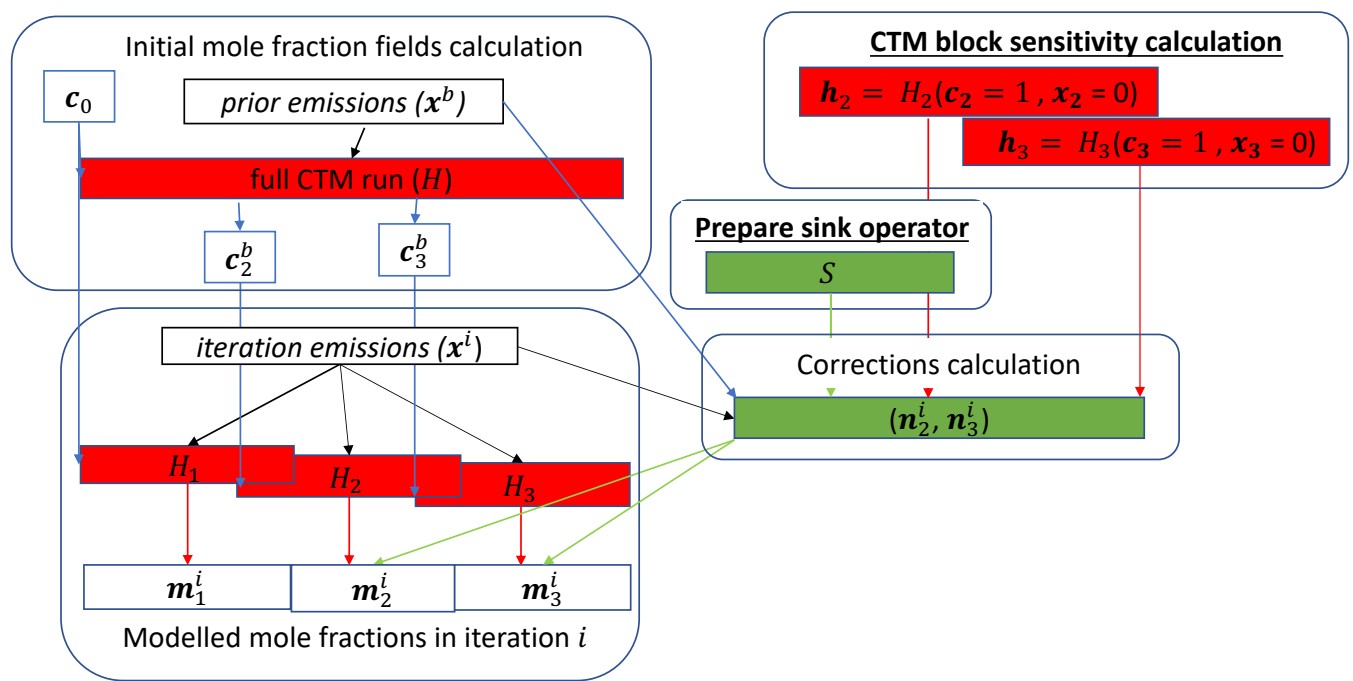

**Figure 1** Schematic diagram of a PP methane inversion's forward mode, which computes mole fractions $m^i$ in iteration $i$. The steps shown with red boxes use CTM runs and take long wall time. The steps shown in green are without CTM runs and require negligible wall time. The subscripts denote the block numbers (except for $c_0$, which is the initial mole fraction field at the start of the inversion). For block 1, the initial mole fraction field ($c_1^b = c_0$) and mole fraction correction vector ($n^i$) is not needed. The overlap between the successive blocks ($H_1, H_2, H_3$) represent the overlap period, where the modelled mole fractions from the preceding block are used in the inversion. The "CTM block sensitivity calculation" and "Prepare sink operator" steps of the PP method are implemented in this study, whereas the rest are from Chevallier (2013). Note that the diagram illustrates the PP splitting into only three blocks, whereas more blocks are used in practice depending on the inversion period.

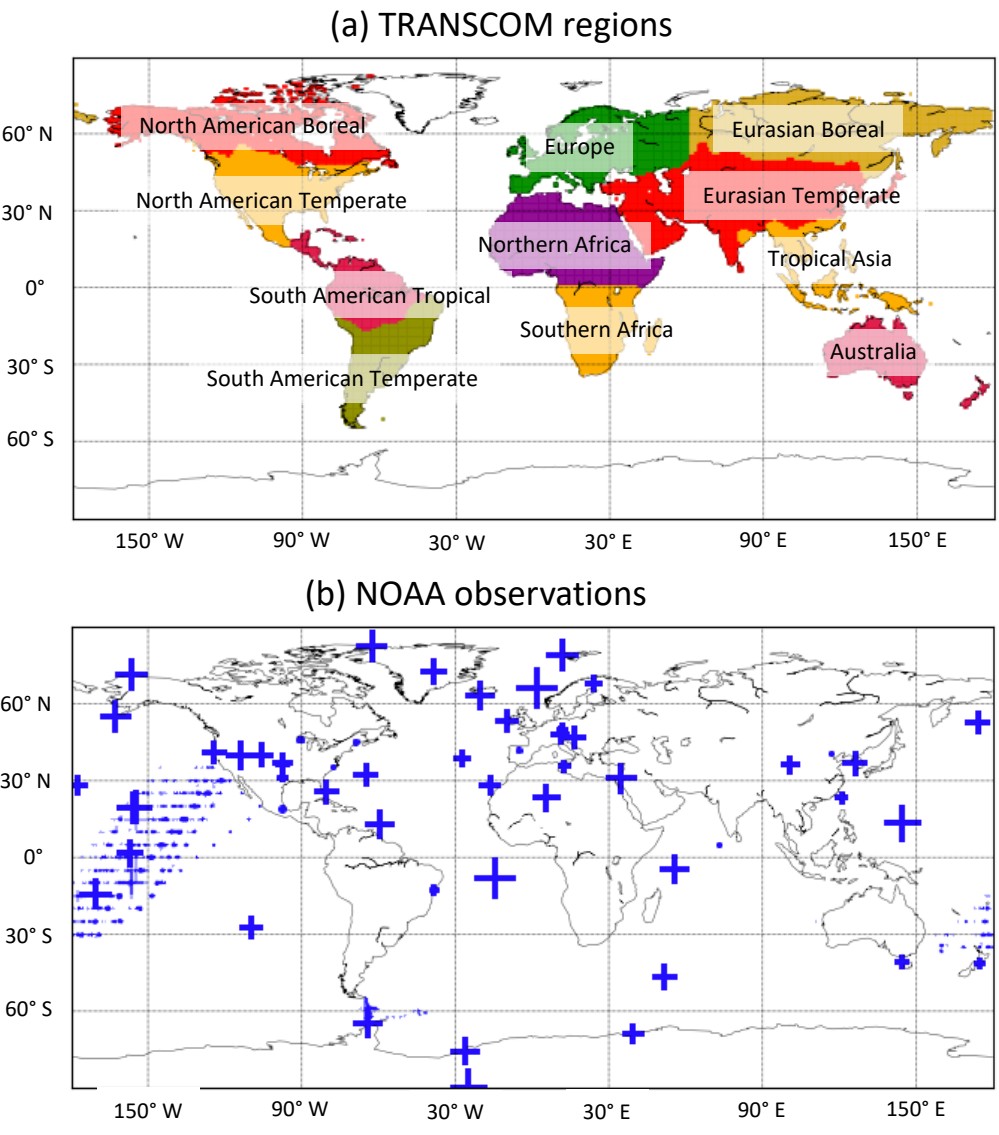

**Figure 2** (a) Definition of the TRANSCOM regions (Gurney et al., 2002). (b) Locations of NOAA methane observation sites used in this study. The size of the symbol "+" is proportional to the number of observations assimilated from each site.

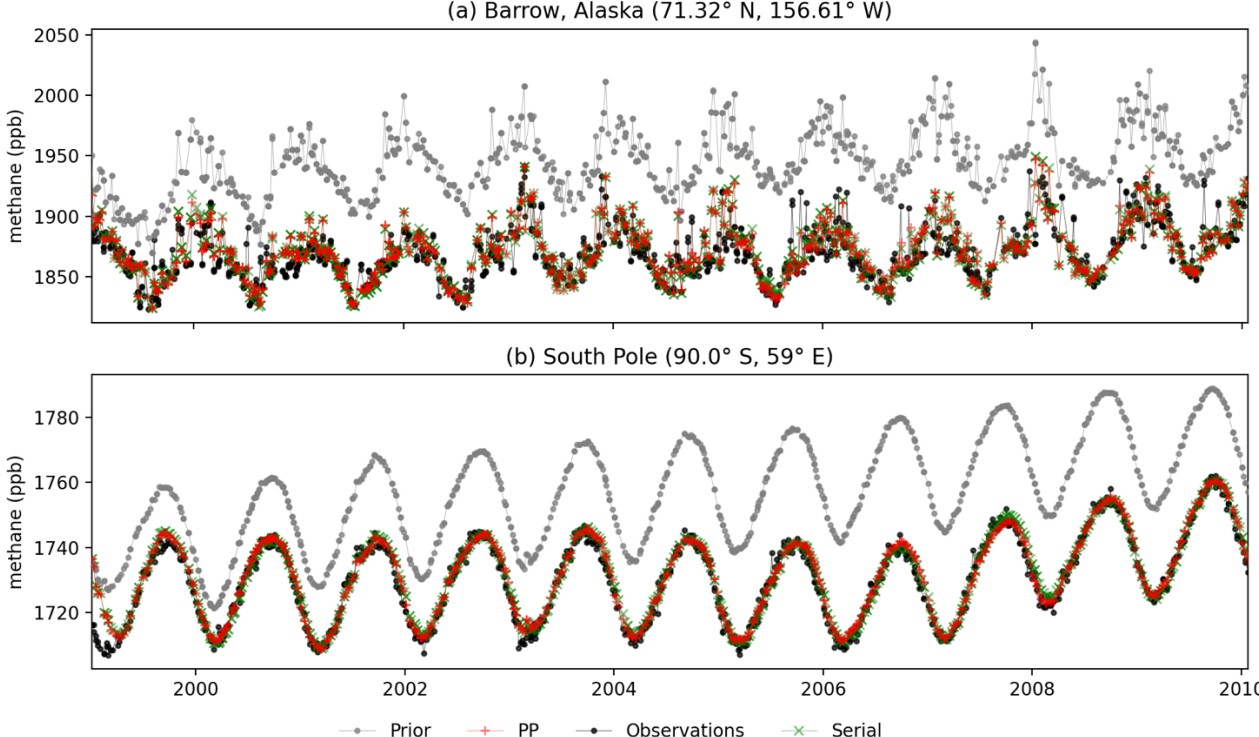

**Figure 3** Modelled and observed methane mole fractions at the two remote background NOAA stations. Barrow, Alaska is shown in panel (a), and the South Pole is shown in panel (b).

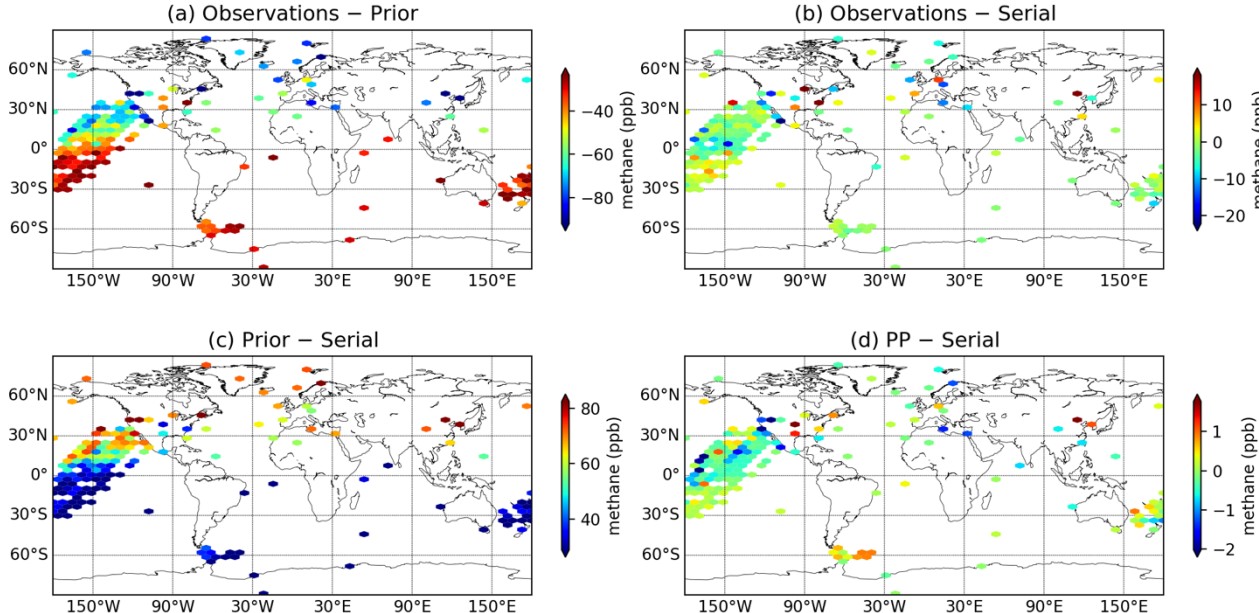

**Figure 4** Methane mole fraction differences at the observation sites (see Figure 2.b). Panel (a), (b), (c) and (d) show the average difference between observations and prior, observation and serial, prior and serial, and PP and serial, respectively. The color scale range is set at mean ± 1 standard deviation of the plotted values.

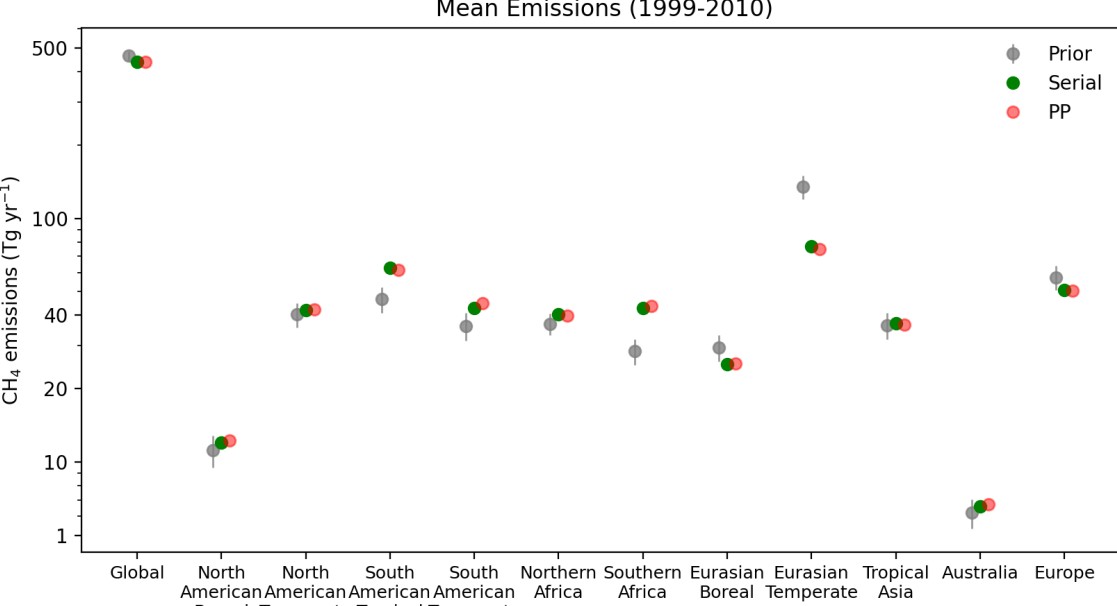


**Figure 5** Total methane emission estimates from the inversions for the globe and TRANSCOM regions (see Figure 2), averaged over 1999-2010. The vertical lines on the markers show the ±2σ uncertainties of the prior emissions.

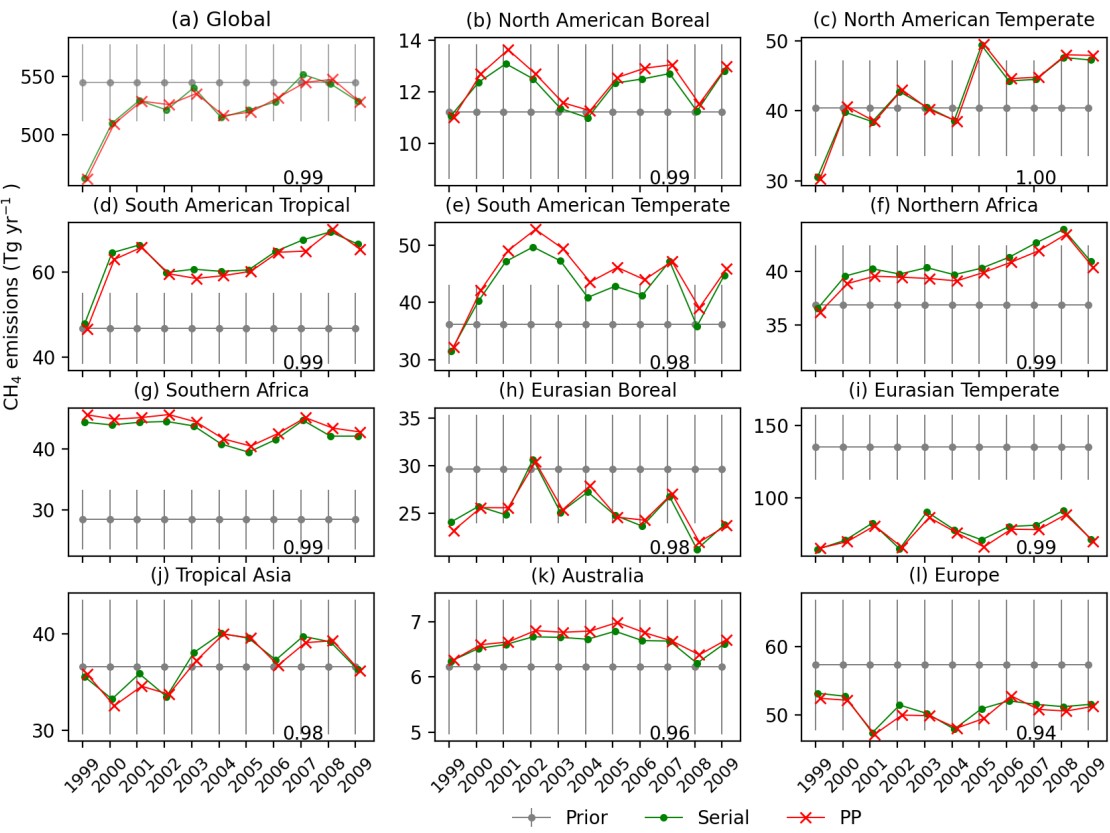

**Figure 6** Annual methane emission estimates from the PP and serial inversions for the globe and the TRANSCOM regions. The vertical bars show the ±2σ uncertainties of the prior emissions. The correlation coefficients of PP and serial time series are given at the bottom of each panel.

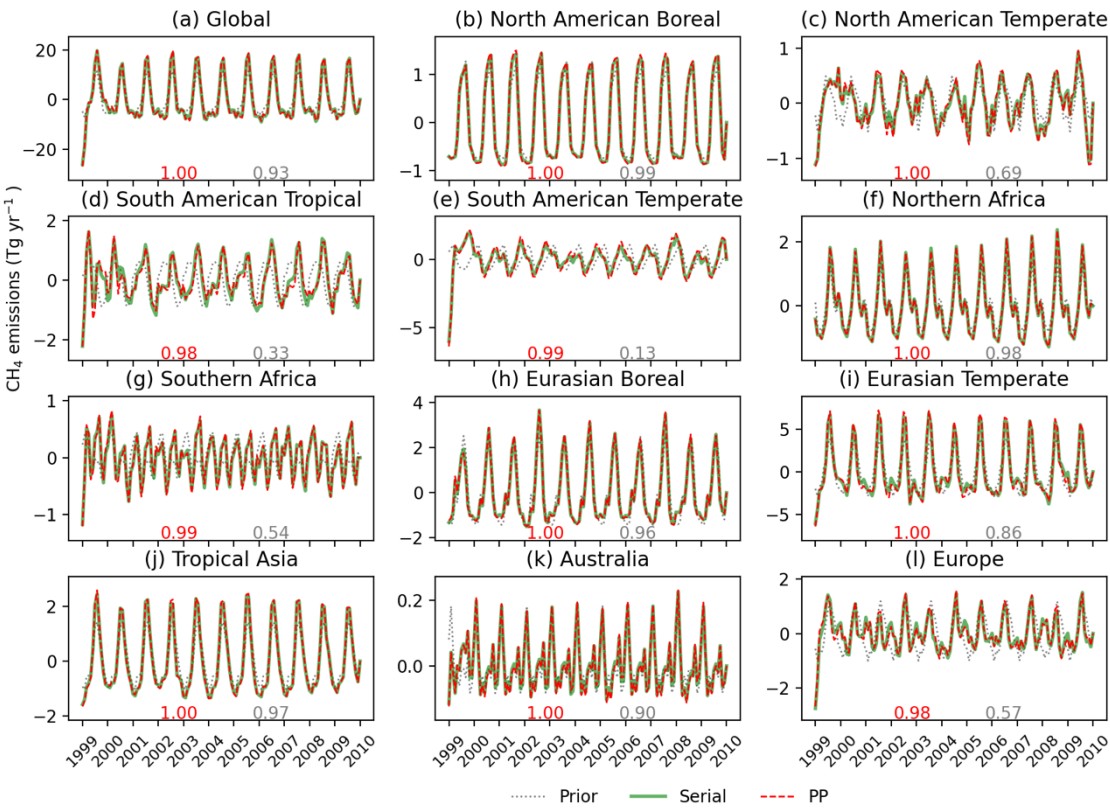

**Figure 7** Intra-annual variation of the PP and serial emissions for the TRANSCOM regions. The correlation coefficients of the PP (red) and prior (grey) time series with the serial time series are given at the bottom of each panel.


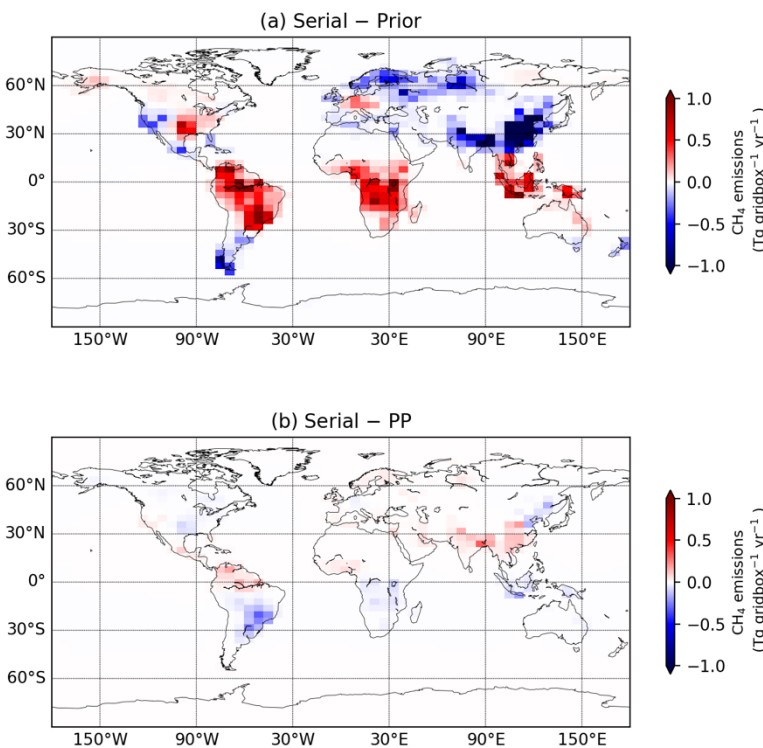

**Figure 8** Emission differences averaged over 1999-2010. Panel (a) shows differences between serial and prior. Panel (b) shows differences between serial and PP.
