# Peer review of "Order of magnitude wall time improvement of variational methane inversions by physical parallelization: a demonstration using TM5-4DVAR"

_Geoscientific Model Development, 2021_

## Referee Comment (RC2)

Review of Pandy et al., gmd-2021-339

**General comments**
This manuscript presents a new method that employs a parallel operation of 4DVAR for the application of atmospheric methane inversions. The original development is done for $CO_2$ application, but this study takes care of atmospheric lifetime in addition. The basic idea is to split the simulation time and run that split blocks in parallel to reduce computational time. This concept itself is not new, and becoming increasingly popular today as required simulation years lengthens. Such development is also important for near real-time understanding of greenhouse gas budgets. There are various ways to deal with the error in the initial condition of each block, and the presented method is scientifically sounding and applicable in other 4DVAR models. The work suites the scope of the journal, and is generally well presented. However, I would like to have some additional explanations and clarifications before publication.

- Study design: This study not only examines the computational performance, but also the accuracy they archived from the new method. Such accuracy would be more precisely examined using perturbed observations from the sequential simulation, i.e. "truth" you aimed to archive at. Was there any specific reason why you did not do so, but used real-life mole fraction data? When using real-life data, exactly the same set of observations should be assimilated, but that information was unclear from the manuscript.

- Calculation of uncertainty: PPVI introduces additional uncertainty to the model estimates, but also probably reduces computational time to calculate flux uncertainties. Please add how uncertainty will be calculated from PPVI properly taking care of the uncertainty in the correction factors. If that is impossible or still under development, please at least add discussion on this point. Would you in future inform uncertainty from those estimates e.g. for all simulation years?

- Consequence to the point above, the study examined only for cases where initial concentration fields are at equal grid resolution to the actual runs. However, the application is meant to run for higher resolution inversion as well. I would not ask to do additional inversion, but can you speculate from e.g. previous studies how it affects the flux errors? For example, you used a single correction factor ($n^i_k$) for each iteration/block globally and annually, but is it a good assumption?

- Introduction: I would like to have more background information on the block-based method. The authors present Chevallier (2013) for the bases of this study, but is it the only study that employs the block-based methods? How do the atmospheric methane inversions generally generate/correct initial conditions? This information would give further insight into what is new in this method.

- As I understood, the method is not to improve the calculation accuracy, but the reduce computational time yet achieving some accuracy at annual/regional levels. Please add in the Conclusion how widely you recommend this method to be used. The authors mention that CAMS simulations will be replaced by this method, but do you see other applications? Can we use those for a more detailed analysis of temporal/spatial distributions? Is this method also applicable in other inversion methods than 4DVAR?

- Additional figures/tables: I would like to see more details about the performances. Please consider including additional information.

- ○ More details on computational time. How much clock time did each step take? As I understood, Table 1 presents the total clock time, but would like to see that for each step, and not only the totals.
- ○ Mole fraction differences in a map. Figure 3 presents the representative sites from the SH and NH, but I would like to see the spatial distribution in more detail.

**Specific comments**

Please ensure that all terms are precisely defined, and not used interchangeably. The term "correction factors" is the one to pay special attention to. You have several correction factors, and were sometimes confusing.

P4 L115: Where the "emission to mole fraction conversion factor $f = 0.361$ ppb/Tg" came from? Please give information on how it was driven.

P5 L147: Please specify which of those steps can be done in parallel (i.e. independent of each other), and which of those steps are needed to re-do (update) for any new inversion runs with changes in inputs, chemistry or years.

P6 L172-174: Did you do any preprocessing of the data? Please also see general comment on point about the observations used.

P6 L181: How do you decide on "9 months"? Do I understand correct that you have 9 months of overlap, i.e. 4.5 months before and after the year-in-question?

P8 L 230: "a hypothetical 35-year inversions using the TM5-4DVAR setup."
What is this?

**Technical comments**

Please check some minor English language errors and technical typos.

Equations and notations therein. Please ensure that the vectors and matrices are in the bold fonts, and add vector/matrix sizes in the explanations. Please also check Figure 1 on this point.

Section 3.1: Please add coordinates of the sites.

Units: Please make sure that units are properly presented in text (e.g. Section 3.1 "The observation-prior mismatch is -6.7 ± 6," → The observation-prior mismatch is -6.7 ± 6 ppb?).

Figure 2: Please present the latitude/longitude units in N/E/S/W, i.e. 60°N instead of +60°.

Figure 4: Please add units to the x-axis.

Figure5: I assume this is regional total emissions, averaged over 1999-2010. Please consider rephrasing the caption.

---

## Author Comment (AC2)

We thank the reviewers for their detailed comments. These comments have led to a significant improvement in the quality and presentation of our manuscript. Our pointwise responses (AR) to reviewers' comments (RC: in *Italics*) and the respective changes in the manuscript (blue text) are as follows:

**Anonymous Referee #1**

*RC: The authors present an implementation of a so-called physical parallelization for variational flux inversions (PPVI): from a previously described PPVI aimed at carbon dioxide ($CO_2$), they add developments to take into account the chemical reactivity of methane ($CH_4$).*

**General comments**

*RC: The developments described in this paper are particularly relevant since long-term methane inversions are*
*now run by several teams and the issue of the trends in methane emissions by various types of sources is still under study. Nevertheless, I think the presentation is too sloppy as it is: the work must be better introduced and described. technically, several notations are unclear in the mathematical description. Moreover, although I am not an native English speaker, I think the writing has to be improved.*

**AR1**: Following the reviewer's suggestions, we have carefully examined these issues and made the necessary
corrections. We have adapted the introduction section (see AR2). The corrections made to the technical notations are given in AR4. Furthermore, we have improved the writing and presentation throughout the manuscript.

*RC: The introduction to the paper is off the mark. It remains very general and not precise enough on variational inversion. Some examples: the state vector, in most inversions, not only consists in emissions but also includes initial conditions or boundary conditions for area-limited domains; the analytical approach is alluded to*
*compared to the variational one but it is never explicitly stated that the analytical approach cannot be used for non- linear problems (which may be the case with reactive species); conversely, it is not stated that the variational approach does not provide full posterior uncertainties as a by-product of an inversion (either none are obtained, or truncated ones). I think the introduction does not target the right readers: people who may be interested in PPVI already know the whys and hows of analytical and variational inversion. It would be more*
*useful to clearly state in which cases and why this implementation of PPVI is interesting e.g. for variational inversions of reactive species at scales at which chemistry is to be taken into account but the precision is not so important i.e. not for non-linear chemistry.*

**AR2**: We have made the introduction section more precise by removing and rearranging the text, and adding the following:

"Trace gas inversions adjust a state vector, which includes gridded emissions (and sometimes initial mole fraction field and other parameters), to improve the agreement between model simulations and observations"

"The variational inversion approach was introduced to lift the state vector size restriction (Chevallier et al., 2005). In this approach, the cost function minimum is computed using an iterative procedure, with each iteration comprising of a forward and an adjoint CTM run. The method has the advantage over the analytical approach in
that it can be applied to non-linear inverse problems. Truncated posterior uncertainties can be obtained from variational inversions using the conjugate gradient minimizer for linear inverse problems (Meirink et al., 2008). A more robust, but computationally expensive, estimate of posterior uncertainties can be obtained using a Monte Carlo method (Chevallier et al., 2007, Pandey et al., 2016). However, as each iteration of a variational inversion uses the output of the previous iteration, the calculations can take months depending on the spatial and temporal
resolution.

**AR3**: The analytical inversion approach was used in Maasakkers et al. (2019) and Zhang et al. (2021) to optimize methane and OH by assuming a quasi-linearity. In the discussion section of the manuscript, we have described the cases where PPVI is useful.

"The utility of the PP method for inversion of a trace gas depends on the time scale of the influence of emissions
on observations within the spatial domain of the CTM. Therefore, PP is mainly useful in global inversions of trace gases that have atmospheric lifetime of a year or longer in the atmosphere. For a trace gas with a shorter lifetime, such as of carbon monoxide with 2 months lifetime, emission perturbations last for a short duration. A multidecadal inversion of such a trace gas can be broken into many short inversions. These short inversions can be performed in parallel, and the posterior emission can be combined thereafter. A similar approach can be used
for regional inversions of short-and long-lived trace gases because emission perturbations are quickly advected out of the regional CTM domain and hence do not influence observations for a long period."

*RC: In Section 2 Physical parallelization for variational inversions, it must be made very clear which parts are the general or Chevallier (2013) developments and which are specific to this work and therefore, to methane. It*
*should make it possible to understand whether the developments are also applicable to other species (e.g. CO). A discussion on the assumptions required to apply this PPVI and its limitations is necessary, either in this part or in the introduction or in the discussion.*

**AR4**: The utility the PP for CO inversions is discussed in AR3. We have clarified which parts of the PPVI approach are from Chevallier (2013) and what we have developed in Figure 1 and text.
"$x^a$. CTM block $H_k$ computes the mole fraction vector $m_k^i$ with emissions $x_k^i$ and mass correction vector $n_k^i$.

$$m_k^i = H_k\left(c_k^a, x_k^i\right) + n_k^i \quad \ldots\ldots\ldots(4).$$

$n_k^i$ accounts for the emission changes from prior in the preceding block. An overlap period between consecutive blocks, where modelled mole fractions from the succeeding block are discarded, is used to distribute the emission changes over the spatial domain of the CTM. The PP method by Chevallier (2013) was applied to $CO_2$ inversions, where $n_k^i$ was simply calculated as the sum of emission changes:

$$n_k^i = \sum_{l=1}^{k-1} f\left[x_l^i - x_l^a\right] \quad \ldots\ldots\ldots (5),$$

where $f$ is a scalar used to convert emissions to mole fractions. $f$ is calculated for a tracer by assuming a uniform distribution of each emission adjustment throughout the Earth's atmosphere.

Methane has an atmospheric lifetime of 10 years. Unlike $CO_2$, a large fraction of a methane emission perturbation will be chemically removed within the duration of a multidecadal inversion. Therefore, in our new implementation of the PP, we modify $\boldsymbol{n}_k^i$ to account for the lifetime by implementing an atmospheric sink operator $S$. In addition, we implement a CTM block sensitivity operator $\mathbf{B}_k$, which distributes emission changes more precisely, as per the atmospheric transport rather than assuming a globally uniform distribution. $\mathbf{B}_k$ is computed at the start of the inversion by running block $H_k$ in forward mode with a unit initial mole fraction field and zero emissions. As $H_k$ is strictly linear for methane, $\mathbf{B}_k$ can be stored as a matrix.

$$\boldsymbol{n}_k^i = \mathbf{B}_k \sum_{l=1}^{k-1} S_{l,k}(f[\boldsymbol{x}_l^i - \boldsymbol{x}_l^a]). \quad \ldots\ldots\ldots\ldots(6). \text{ ''}$$

"

**Figure 1** Schematic diagram of a PP methane inversion's forward mode, which computes mole fractions $\boldsymbol{m}^i$ in iteration $i$. The subscripts denote the block numbers (except for $\boldsymbol{c_0}$, which is the initial mole fraction field at the start of the inversion). For block 1, the initial mole fraction field ($\boldsymbol{c}_1^a = \boldsymbol{c_0}$) and mole fraction correction vector ($\boldsymbol{n}^i$) is not needed. The overlap between the successive blocks ($H_1, H_2, H_3$) represent the overlap period, where the modelled mole fractions from the preceding block are used in the inversion. The "CTM block sensitivity calculation" and "Prepare sink operator" steps of the PP method are implemented in this study, whereas the rest are from Chevallier (2013)."

*RC: In Section 3PPVI Performance test, not all the information required to understand (and reproduce) the simulations are available. The main information missing is how the posterior uncertainties are obtained: which approach is used? What are the assumptions? Even the simple approach of using Congrad as a minimizer and using the uncertainties obtained with a truncation requires to specify at least this truncation threshold and how it is expected to affect the resulting uncertainties estimates.*

**AR5:** We have added the following text to the manuscript to provide more information on our test setup:

"Both inversions are performed for an 11-year period (1999-2010) with identical observations and prior
emissions. …..

……… The meteorological fields for this offline model are taken from the European Centre for Medium-Range
Weather Forecasts (ECMWF) ERA-Interim reanalysis (Dee et al., 2011).

…. The cost function $J$ is minimized using the conjugate gradient minimizer, which is based on the Lanczos
algorithm (Fisher and Courtier, 1995). The inversions use the convergence criterion of gradient norm reduction
by a factor 1000, which is achieved after 19 iterations in both inversions."

**Specific comments**

*- Section 2 Physical parallelization for variational inversions:*

**RC:** *p.3 l.93 in Eq.2: it should be $H^*$ and not $H^T$ - or the assumptions which make $H^T$ equal to $H^*$ should be*
*stated. It would also be safer to add a bracket: $H^* [ R^{-1}(H(x^i)-y) ]$*

**AR6:** Done.

**RC:** *p.3 l.96: same remark as above: $H^*$ is the adjoint, if $H^T$ is used, it means that the problem is linear, which*
*must be stated explicitly from the beginning.*

**AR7**: Done.

**RC:** *p.4 l.116: why this conversion factor?*

**AC8**: We have added the following text to answer:

"$f$ is a scalar used to convert emissions to mole fractions assuming a uniform distribution of the emitted trace gas
throughout the Earth's atmosphere. f is calculated simply as the ratio between the number of moles in a unit
emission and the number of moles of air in the atmosphere."

**RC:** *p.4 l.119 in Eq. 5: the notation for H changes suddenly from italics i.e. an operator with no particular*
*characteristics to bold i.e. a matrix (probably): see above for the issue about H and its various spin-offs being*
*linear or not and adjust notations accordingly.*

**AR9**: Done.

**RC:** *p.5 l.134 in Eq 6 and seq.: the notation * for the adjoint appears here: please make this consistent with the beginning of the Section. Moreover, H is bold so probably a matrix i.e. for a linear problem so that * and transpose are the same: this is not clear at all for the reader.*

**AR10:** Done.

*- Section 3 PPVI performance test*

**RC:** *p.7 l.210-211: the difference between both posterior emissions must be compared to the difference with the prior to be said to be small - or not. Better still, the uncertainties on the three estimates must be taken into account for such a comparison.*

*p.7 l.213-214: same remark as above for the regions: how much is the deviation from the prior compared to the*
*5% between the two inversions? What about the uncertainties on the emission estimates?*

**AR11:** See our reply on the issue of uncertainties in AR12. We have added a comparison with the prior emission adjustments:

"The serial inversion adjusts the global mean prior emissions of $544 \pm 11$ Tg yr$^{-1}$ by $-22$ Tg yr$^{-1}$. The PP inversion is in excellent agreement with the serial inversion in this respect. The two differ by  0.3 Tg yr$^{-1}$
$^1$ (0.06%), which is 1% of the difference between prior and posterior emissions from the serial inversion. The global methane emissions are in general well constrained by the NOAA observations in a serial inversion, and the additional error introduced by the PP  method does not seem to have a significant impact on the global emissions. At regional scales, the serial inversion adjustment is the smallest for Australia: $+ 0.4$ Tg yr$^{-1}$ for a prior of $6.6 \pm 0.4$ Tg yr$^{-1}$. The PP inversion adjusts the prior
here by 0.5 Tg yr$^{-1}$, implying that the difference with the serial inversion (0.1 Tg yr$^{-1}$) is well within the prior emission uncertainty. The serial inversion changes the Eurasian temperate emissions the most, by $-58$ Tg yr$^{-1}$, where prior emissions are $135 \pm 8$ Tg yr$^{-1}$. The PP inversion changes these emissions by $-60$ Tg yr$^{-1}$, i. e., a difference of 2 Tg yr$^{-1}$ and well within the prior uncertainty also. The South American temperate region has the largest difference between the serial and PP emission estimates of 2 Tg yr$^{-1}$.
The serial emissions for this region are 6.5 Tg yr$^{-1}$ higher than the prior of $36 \pm 2.4$ Tg yr$^{-1}$.

**RC:** *p.7 l.214: the posterior uncertainty is alluded to here but nowhere is it stated how it is computed. Since the full posterior uncertainties are not a by-product of the variational inversion, the way they are computed must be described (truncated from Congrad? ensemble? Monte-Carlo? other method?).*

**AR12:** The posterior uncertainties presented in this study are truncated estimates derived using a limited number (number of iterations) of eigenvectors and eigenvalues of the Hessian, which are obtained from the conjugate gradient minimizer. The truncated posterior uncertainties are, by definition, an overestimate. We have realized that assessment in PP and serial posterior differences against those uncertainties is an unfair test. Therefore, we have removed the posterior uncertainties from the revised manuscript. We now evaluate the error in PP emissions
against the differences between the prior and the serial inversion (AR11). The ability of the PP inversion to recover the serial uncertainties is important. However, a much larger number of inversion iterations than preformed in this study are needed for the Hessian eigen values to converge. This is especially difficult for the serial inversions. The assessment of the posterior uncertainties of the PP method will be performed in a future study.

**RC:** *p.7 l.215 seq.: I guess the correlation coefficient used here is simply the correlation of the time series. There are other characteristics of the inter-annual variability which could be interesting to look at e.g. are the uncertainties the same?*

**AR13:** See AR12 for our response about uncertainties. The difference between the 1999-2010 averages and correlation of the inter- and intra-annual (now added to the manuscript) variations time series provides a good comparison of the posterior emission estimates. We have added a new figure to present the seasonal and intra-annual variation of the emission time series:

"

[Figure]

Figure 7 Intra-annual variation of the PP and serial emissions for the TRANSCOM regions. The correlation coefficients of the PP (red) and prior (grey) time series with the serial time series are given at the bottom of each panel"

We have added the following text to discuss this figure in Section 3.1:

"Figure 7 shows the intra-annual variations of the emissions. At the global scale, the PP and serial time series match very well with R = 1.00, whereas R between prior and serial is 0.93. The agreement between PP and serial
time series is also very good for all TRANSCOM regions (R > 0.98) despite low correlations between prior and serial emissions for some regions, for example, R = 0.13 for the South American temperate region. This shows that the PP inversion is able to reproduce the seasonal cycle of the emissions very well. In summary, the combination of small differences in the mean emissions, and the high correlations between intra- and inter-annual time series, shows that the PP inversion can effectively reproduce results of the serial inversion."

**RC:** *p.8 l.233-234: what about the uncertainties? Without an explanation on how they are computed, the times given here are read as times for one inversion and may therefore be a lot smaller than what is actually required to get the full range of results (i.e. emission estimates + uncertainties).*

**AR14:** A large number of iterations are needed to get an reasonable estimate of the uncertainties with the
conjugate gradient method, for both the serial and the PPVI inversions. This is especially very difficult for the serial inversion. We have discussed this issue in more detail AR12. We have modified the sentence in the revised manuscript:

"Overall, we find that the PP method, which accounts for the atmospheric lifetime of methane, is able to effectively reproduce the posterior emissions of a traditional 11-year serial inversion 5 times faster."

**RC:** *p.9 l.263 seq.: the specification of the OH fields is one of the main issues for methane inversions today, particularly as the vertical distribution of OH is crucial when using satellite data. A sink defined as simply as suggested here (even with an annual change) does not really solve the scientific issue. The optimization of the sink, as described in Section 4.3 is one of the possible ways forward.*

**AR15:** Depending on the scientific question, there are multiple issues with methane inversions today, including
transport model errors and specification of OH fields. A wrong OH specification in a surface data-only inversion will also cause an error in results. Satellite data can better constrain OH. The topic of which method of OH specification in methane inversions is better is outside the scope of this study. We have discussed in the manuscript how OH can be simultaneously optimized within the PP framework in the discussion section.

*- Figure 1*

**RC:** *It would be useful to distinguish between the general ($CO_2$) PPVI and the elements which are particular to this work i.e. the $CH_4$. for example, the sink does not appear in this figure. Please also check the consistency of the notations (matrices, operators, vectors,...).*

**AR16:** Done. See AR4.

**RC:** *- Figures 5, 6 and 7. How are the uncertainties obtained? Does a 2-sigma interval make sense?*

**AR17:** See AR12.

***Technical corrections***

**RC:** *Throughout the text, "a priori" and "a posteriori" are used: shouldn't it be "prior" and "posterior" instead?*

*There are many writing mistakes, such as sentences where words are missing (e.g. p.7 l.194: "the PPVI results are good agreement with the results from serial") or superfluous words remain: the text must be re-read carefully*
*by the authors before being checked by a native speaker.*

**AR18:** Thank you for pointing out these issues. We have corrected the mistakes.

*Review of Pandy et al., gmd-2021-339*

***General comments***

**RC:** *This manuscript presents a new method that employs a parallel operation of 4DVAR for the application of atmospheric methane inversions. The original development is done for CO2 application, but this study takes care of atmospheric lifetime in addition. The basic idea is to split the simulation time and run that split blocks in*
*parallel to reduce computational time. This concept itself is not new, and becoming increasingly popular today as required simulation years lengthens. Such development is also important for near real-time understanding of greenhouse gas budgets. There are various ways to deal with the error in the initial condition of each block, and the presented method is scientifically sounding and applicable in other 4DVAR models. The work suites the scope of the journal, and is generally well presented. However, I would like to have some additional explanations and*
*clarifications before publication.*

- *Study design: This study not only examines the computational performance, but also the accuracy they archived from the new method. Such accuracy would be more precisely examined using perturbed observations from the sequential simulation, i.e. "truth" you aimed to archive at. Was there any specific reason why you did not do so, but used real-life mole fraction data? When using real-life data, exactly*
*the same set of observations should be assimilated, but that information was unclear from the manuscript.*

**AR19:** Our study shows that the PP method can compute results comparable to a serial variational inversion in a fraction of wall time. The PP method mainly reduces the CTM run time in an iteration of the inversion. Hence, the benchmark for the PP method's performance is simply a real-world serial inversion. A synthetic inversion test
setup with pseudo-observations would not have all the complexities of a real-world inversion (for example, transport model biases). Therefore it is not a better test of the PP method. We have used the same set of observations in the two inversions. See AR5.

**RC:** *Calculation of uncertainty: PPVI introduces additional uncertainty to the model estimates, but also*
*probably reduces computational time to calculate flux uncertainties. Please add how uncertainty will be calculated from PPVI properly taking care of the uncertainty in the correction factors. If that is impossible or still under development, please at least add discussion on this point. Would you in future inform uncertainty from those estimates e.g. for all simulation years?*

**AR20:** The PP method reduces the wall time of an inversion iteration. Therefore the wall time to calculate flux
uncertainties is also reduced. This applies to both methods of variational inversion's uncertainty calculation: the Monte Carlo method or Hessian approximation from the conjugate gradient minimizer. Accounting for the additional uncertainty due to PP will be part of future developments. We have added text to the discussion section to elaborate on this:

"The PP method reduced the wall time of the CTM simulations in a variational inversion but introduces additional model errors because of the simplifications made. For our test inversion setup, these PP-CTM model errors are minor as the posterior PP emission estimates are in good agreement with the serial estimates. In future PP implementations, these PP-CTM errors can be accounted for in the observation error matrix **R**. The PP-CTM error can be calculated as the difference between the model output of a PP and a serial forward CTM run with randomly perturbed prior emissions."

**RC:** *Consequence to the point above, the study examined only for cases where initial concentration fields are at equal grid resolution to the actual runs. However, the application is meant to run for higher resolution inversion as well. I would not ask to do additional inversion, but can you speculate from e.g. previous studies how it affects the flux errors? For example, you used a single correction factor ($n^i k$) for each iteration/block globally and annually, but is it a good assumption?*

**AR21:** The mole fraction corrections $\boldsymbol{n}_k^i$ is a vector of size of the observations vector within a block. We do not expect the performance of PP to degrade significantly when inversion is performed at a higher spatial resolution given there is a sufficient overlap period and the $\boldsymbol{n}_k^i$ is correctly parameterized. Our performance tests shows that a single correction for the whole globe performs well with a 9-month overlap between the blocks. The overlap period can be reduced to save more time and computational resources. This would require multiple mole fraction
corrections calculations, for example, one for each hemisphere. We have added the following text to the revised manuscript:

       "The PP accuracy could be maintained with shorter overlap periods by using a mole fraction correction vector per hemisphere rather than the single global vector used in this study. However, the computational resources and wall time saved by this would be partially spent on the additional block sensitivity runs. Our test inversions are
performed at a relatively coarse horizontal resolution of 6° × 4° with 25 vertical hybrid sigma-pressure levels. We do not expect the performance of the PP method to degrade significantly for higher resolution inversions if there is sufficient overlap between the blocks and the mole fraction corrections are parameterized correctly. Furthermore, the performance gained by performing the inversions at higher resolution because of the improved computational performance will likely outweigh the accuracy loss due to the assumptions made in the PP
method."

       **RC:** *Introduction: I would like to have more background information on the block-based method. The authors present Chevallier (2013) for the bases of this study, but is it the only study that employs the block-based methods? How do the atmospheric methane inversions generally generate/correct initial conditions? This information would give further insight into what is new in this method.*

**AR22:** Besides Chevalier (2013), we are not aware of a study that uses a block-based method to implement physical parallelization on the CTM in an atmospheric inversion. For a serial inversion, an initial concentration field is only needed at the start of the inversion period. Some inversion studies optimize the initial concertation fields within the inversions. Others studies use a spin-up period to remove the impact of initial concentrations errors on posterior emissions.

**RC:** *As I understood, the method is not to improve the calculation accuracy, but the reduce computational time yet achieving some accuracy at annual/regional levels. Please add in the Conclusion how widely you recommend this method to be used. The authors mention that CAMS simulations will be replaced by this method, but do you see other applications? Can we use those for a more detailed analysis of temporal/spatial distributions? Is this method also applicable in other inversion methods than 4DVAR?*

**AR23:** We have described the cases where PPVI is useful in AR3. The PP method can be used for OH optimization in a methane or methyl chloroform inversion (see Section 4.3 of the manuscript). We do not foresee the application of PP to inversion methods other than 4DVAR.

   **RC:** *Additional figures/tables: I would like to see more details about the performances. Please consider including additional information.*

**RC:** *More details on computational time. How much clock time did each step take? As I understood, Table 1 presents the total clock time, but would like to see that for each step, and not only the totals.*

   **AR24:** We have added the following text to the revised manuscript:

"The main steps of PP implementation are listed in Section 2. In our inversion test, the initial mole fraction fields c_0 (step 1) were taken from an inversion using surface measurements that was not performed in this study. Steps
1, 2, 5.a.ii and 6.a.ii took negligible time. Step 3 took 2.5 hours because it consists of a full serial TM5 forward run. Steps 4, 5.a.i and 6.a.i consisting of 11 21-month TM5 simulationsrun over blocks overof 21 months which were run in parallel and, took this 25 minutes each. Note that an iteration took longer than the sum of the forward and adjoint block runs because of a few minutes waiting time for the computer cores to become available again."

**RC:** *Mole fraction differences in a map. Figure 3 presents the representative sites from the SH and NH, but I would like to see the spatial distribution in more detail.*

   **AR25:** We have changed Figure 4 of the manuscript. It now shows the spatial distribution of the mean mismatches at the observation sites.

"

[Figure]

**Figure 4** Methane mole fraction differences at the observation sites (see Figure 2.b). Panel (a), (b), (c) and (d) show the average difference between observations and prior, observation and serial, prior and serial, and PP and serial, respectively. The color scale range is set at mean ± 1 standard deviation of the plotted values."

**AR26:** We have added the following text to discuss the new figure:

"Figure 4 shows the average mole fraction differences at all observation sites. The observation-prior RMSD for all observations combined is 67 ppb. The mean mismatch is –58 ppb because the 2008 bottom-up emissions used as the prior are larger than the mean posterior emission over 1999-2010. The average data uncertainty (mean of the square root of the diagonal elements of R) is 19 ppb (not shown). For both inversions, a good model fit to the observations is achieved with a gradient norm reduction of 1000. The posterior simulation of both the serial and

PP inversions reduce the RMSD to 20 ppb (mean = –2 ppb). The RMSD between PP and serial is 1.9 ppb (mean = –0.1 ppb), which is an order of magnitude smaller than the posterior-prior RMSD of 62 ppb (mean = –55 ppb). This shows that the implementation of the PP method has little impact on the inversion's ability to fit the observations.

**Specific comments**

      **RC:** *Please ensure that all terms are precisely defined, and not used interchangeably. The term "correction factors" is the one to pay special attention to. You have several correction factors, and were sometimes confusing.*

**AR27:** Done

**RC:** *P4 L115: Where the "emission to mole fraction conversion factor f = 0.361 ppb/Tg" came from? Please give information on how it was driven.*

**AR28:** We have added the following text to the revised manuscript:

"$f$ is a scalar used to convert emissions to mole fractions assuming a uniform distribution of the emitted trace gas throughout the Earth's atmosphere. f is calculated simply as the ratio between the number of moles in a unit
emission and the number of moles of air in the atmosphere."

**RC:** *P5 L147: Please specify which of those steps can be done in parallel (i.e. independent of each other), and which of those steps are needed to re-do (update) for any new inversion runs with changes in inputs, chemistry or years.*

**AR29:** Every stated step will need to be redone for any new inversion with changes in inputs, chemistry or years.
We have added the following text to the revised manuscript:

"The CTM runs in the steps 4, 5.a.i and 5.b.i are performed in parallel."

**RC:** *P6 L172-174: Did you do any preprocessing of the data? Please also see general comment on point about the observations used.*

**AR30:** No pre-processing of the data was needed.

**RC:** *P6 L181: How do you decide on "9 months"? Do I understand correct that you have 9 months of overlap, i.e. 4.5 months before and after the year-in-question?*

**AR30:** We have added the following text to the revised manuscript:

"The first 9 months of each block is the overlap period used for uniformly mixing the emission changes within the atmosphere, while the last 12 months provide modelled mole fractions for assimilating the observations."

"The 6-month overlap used by Chevalier et al. (2013) for CO2 inversion was found to be insufficient for a PP methane inversion, likely because of the differences between the source and sink distributions of methane and
CO2. Increasing the overlap period to 9-month and using CTM block sensitivity vector solved this issue. We expect that a 1-year overlap, equal to the interhemispheric mixing time, would be more than sufficient for all tracers irrespective of their source-sink distribution and lifetime"

**RC:** *P8 L 230: "a hypothetical 35-year inversions using the TM5-4DVAR setup." What is this?*

**AR31:** We calculate the expected wall time if a 35-year inversion. We have modified the text in the revised manuscript:

"Table 1 also provides a projection of the wall time improvement of a hypothetical 35-year inversion (not performed in this study) based on the TM5-4DVAR inversion setup used in this study"

**Technical comments**

**RC:** *Please check some minor English language errors and technical typos.*

*Equations and notations therein. Please ensure that the vectors and matrices are in the bold fonts, and add vector/matrix sizes in the explanations. Please also check Figure 1 on this point.*

*Section 3.1: Please add coordinates of the sites.*

*Units: Please make sure that units are properly presented in text (e.g. Section 3.1 "The observation-prior mismatch is -6.7 ± 6," → The observation-prior mismatch is -6.7 ± 6 ppb?).*

*Figure 2: Please present the latitude/longitude units in N/E/S/W, i.e. 60°N instead of +60°. Figure 4: Please add units to the x-axis.*

*Figure5: I assume this is regional total emissions, averaged over 1999-2010. Please consider rephrasing the caption.*

**AR32:** We have implemented the suggested corrections in the revised manuscript.

**RC:** *The authors have developed a kind of "window-splitting" scheme for a variational inverse analysis of atmospheric CH4, which can be performed by parallel computing. For a multi- decadal analysis of long-lived species such as CO2 and CH4, a variational inverse analysis would be time consuming even when a massive amount of computational resources are available. This is because a variational analysis is basically a serial computation algorithm, which requires iterative calculations. In this regard, the developed method is worthy of publication from GMD, though its basic idea is already published by Chevallier (2013). Before publication, however, the reviewer would like the authors to revise the manuscript considering comments described below.*

*It is difficult to follow the description of the scheme, whose major reason is that many matrices and vectors are not written in bold fonts. This is very confusing. Furthermore, the reviewer strongly recommend that the author should clearly describe what is new and different from the original scheme of Chevallier (2013).*

**AR33.** We have corrected the vector and matrix notations and clearly described the new developments on the PP method presented in this study in the text and Figure 1. See AR4.

**RC:** *Although the reviewer is not a native English speaker, the reviewer thinks that the English writing of the manuscript has much room to improve. Therefore, a native check is also recommended.*

**AR34:** We have improved the English writing in the revised manuscript.

**RC:** *The authors claim that the developed scheme is effective for a long-term inverse analysis in terms of wall clack time. The reviewer has no doubt about it, but would like the authors to discuss its relative effectiveness comparing with other approaches. For instance, a MPI parallelization (much more scalable parallelization than OpenMP) on the transport model could also shorten the wall clack time.*

**AR35:** MPI and OpenMP reduce the clock time of transport models by parallelizing the code of the CTM. The physical parallelization methods presented here and in Chevallier (2013) are different because they improve the wall clock time of a variational inversion for any given speed of the transport model.

***Specific comments:***
**RC:** *L11: "variational (4DVAR)" => "four-dimensional variational (4DVAR)"*

**AR36:** Done.

**RC:** *L21: "by a factor of 5" its computational effectiveness should be also described. How much computational resources are increased?*

**AR37:** We have added the following text to give the requested information:

"Note that although the PP inversion took a shorter wall time, it needed extra CPU core hours for the additional 9-month overlap, CTM block sensitivity and initial mole fraction computation runs. The PP inversion used a total of 700 CPU core hours, whereas the serial inversion used about 400 CPU core hours."

**RC:** *L25: "CAMS (Copernicus Atmosphere Monitoring Service)" => "Copernicus Atmosphere Monitoring Service (CAMS)" L39: "CTM (chemical transport model)" => "a chemical transport model (CTM)" L42: Which is "this study", the study by the authors or the one by Saunois et al.? Maybe it is the latter, but it should be clarified for more general readers.*

**AR38:** All done.

**RC:** *L56: "representing the sensitivities by a statistical ensemble" is not clear.*

**AR39:** We have modified the text to clarify:

"The ensemble approach improves the computational performance by parameterizing the state vector sensitivities using a statistical ensemble (Peters et al., 2005)"

**RC:** *L61: "is obtained" would be better than "is computed"*

**AR40:** Done.

**RC:** *L67: "computational efficiency" might be inappropriate, because the computational resources used in the inversion were increased.*

**AR41:** We have changed it to "wall time".

**RC:** *L76: Chevallier (2013) named the scheme as "physical parallelization (PP)", but the authors here named their scheme as "physical parallelization for variational inversion (PPVI)". Are they the same? If that is the case, it would be better to use PP rather than PPVI to respect the original idea of Chevallier (2013).*

**AR42:** We have replaced "PPVI" with "PP".

**RC:** *Somewhere in Introduction: More introduction about CH4 inverse analyses other than Saunois et al. (2020) would be beneficial.*

**AR43:** The PP method is only applicable to long-duration/multidecadal methane inversions. A set of nine such inversions are presented in Saunois et al. (2020). Therefore, we refer the reader to Saunois et al., (2020) to learn about the results of multidecadal methane inversions. Following the suggestion of the first reviewer, we have made the introduction more precise (See AR2). Therefore we have not added more information about other methane inversions.

**RC:** *L87: transpose "T" is missing. "$(x-x_a)B^{-1}(x-x_a)$" => "$(x-x_a)^T B^{-1}(x-x_a)$", "$(H(x) - y)R^{-1}(H(x) - y)$" => "$(H(x) - y)^T R^{-1}(H(x) - y)$". L89 and elsewhere: "In here" => "Here" L89: "the a" => "the"*

**AR44:** All done

**RC:** *L118: Why can the CTM that calculates the initial mole fraction fields be performed at the coarser*
*resolution?*

**AR45:** We have added the following text to answer this:

"Step 3 is the most time-consuming because a full serial CTM run is performed in the step. To reduce the wall time, this run can be performed at a coarse CTM resolution. This will not have a major impact on the inversion's performance as the coarse resolution mole fraction fields would be consistent with the source, sink and large-
scale atmospheric transport patterns, and $m$ is sampled after the coarse field is transported by a high-resolution CTM block runs during the overlap periods."

**RC:** *L119: What is the "methane perturbation"?*

**AR46:** Methane perturbation are methane emission differences between the prior and emissions in an iteration. We have clarified it in the revised manuscript.

"Here the scalar $n_k^i$ accounts for the emission differences between the prior and the iteration in the period preceding the block."

**RC:** *L116: Please describe how the mole fraction conversion factor (=0.361) is derived.*

**AR47:** See AR8

**RC:** *Eqs. (3)-(5): Are $c_0$, $x^i$, $n^i$ scalars or vectors? If they are scalars, are they the global totals?*

**AR48:** All are vectors. They are denoted in bold italics in the revised manuscript.

**RC:** *L124: Please elaborate the sufficiency of the e-folding decay function, because this might be the new and different from the original scheme of Chevallier (2013).*

**AR49:** In Section 4.2, there is more discussion on the e-folding decay function and possible improvement to the
sink operator needed for longer inversions than the PP inversion performed in this study.

**RC:** *L138: What is "the adjoint test"? Please elaborate it.*

**AR50:** We have added the following text to elaborate:

"The correct adjoint implementation of the PP method can be verified using the adjoint test (Meirink et al., 2008). The test checks for the equality

$$\langle M(a), b \rangle = \langle a, M^*(b) \rangle \ \ldots\ldots\ (9),$$

where $M$ and $M^*$ denote the forward and adjoint model operators, $\langle \ \rangle$ denotes the inner product. $\boldsymbol{a}$ and $\boldsymbol{b}$ are the arbitrary forward and adjoint model states."

**RC:** *L152: "uniform" is better than "unity", isn't it?*

**AR51:** Done.

**RC:** *L192-193: Are "78 ppb" and "28 ppb" the results of serial or PPVI?*

**AR52:** It is the RMSD between observation and prior modelled mole fractions. The prior mole fractions are the same for PP and serial.

**RC:** *L203-204: "For both inversions, the good fit .... a gradient reduction of 1000 is sufficient" The fit to the observations cannot be used to determine the sufficiency of the convergence.*

**AR53:** Agreed. We have modified the sentence:

"For both inversions, a good model fit to the observations is achieved with gradient norm reduction of 1000."

**RC:** *L207-208: "The parallelized ... in the serial inversion" is not clear.*

**AR54:** We have modified the sentence to clarify:

"The physically parallelized CTM used in the PP inversion has lost some of the consistency of the full CTM used in the serial inversion."

**RC:** *Section 3.1: One may want to see differences of more small scales (e.g., flux patterns, seasonal cycles).*

**AR55:** We have added another figure and provided a more detailed assessment of the flux patterns. See AR13.

**RC:** *Section 4.1: This section would be better to be moved to Introduction. L257: "if future" => "in future"?*

**AR56:** Reviewer 1 has asked us make the Introduction more precise (see AR2). The explanation of the current CAMS inversion setup in the introduction will be distracting as it is not needed introducing the PP method.

**RC:** *L278: Please spell out "SWIR" and TIR, because they appear first here. L280-281: "These studies ... small in an inversion." Is not clear.*

**AR57:** Done.

**RC:** *L282: Does "the methane lifetimes in the S operator would be scaled in each iteration" mean that S is included in the control variables?*

**AR58:** Yes, we have added more text to clarify:

"Under a quasi-linear assumption, OH can be optimized in a PP methane inversion by introducing annual OH scaling factors in the state vector and the methane lifetimes in the sink operator can be scaled in each iteration to reflect the corresponding OH adjustments."

**References**

Chevallier, F., Breon, F.-M., and Rayner, P. J.: Contribution of the Orbiting Carbon Observatory to the estimation of CO2 sources and sinks: Theoretical study in a variational data assimilation framework, J. Geophys. Res.-Atmos., 112, d09307, https://doi.org/10.1029/2006JD007375, 2007.

Dee, D. P., Uppala, S. M., Simmons, A. J., Berrisford, P., Poli, P., Kobayashi, S., Andrae, U., Balmaseda, M. A.,
Balsamo, G., Bauer, P., Bechtold, P., Beljaars, A. C. M., van de Berg, L., Bidlot, J., Bormann, N., Delsol, C., Dragani, R., Fuentes, M., Geer, A. J., Haimberger, L., Healy, S. B., Hersbach, H., Hólm, E. V., Isaksen, L., Kållberg, P., Köhler, M., Matricardi, M., McNally, A. P., Monge-Sanz, B. M., Morcrette, J.-J., Park, B.-K., Peubey, C., de Rosnay, P., Tavolato, C., Thépaut, J.-N., and Vitart, F.: The ERA-Interim reanalysis: Configuration and performance of the data assimilation system, Q. J. Roy. Meteor. Soc., 137, 553–597, 2011.

Fisher, M. and Courtier, P.: Estimating the covariance matrices of analysis and forecast error in variational data assimilation, in: ECMWF Technical Memorandum 220, ECMWF, Reading, UK,. doi: 10.21957/1dxrasjit, 1995.

Maasakkers, J. D., Jacob, D. J., Sulprizio, M. P., Scarpelli, T. R., Nesser, H., Sheng, J. X., Zhang, Y., Hersher,
M., Anthony Bloom, A., Bowman, K. W., Worden, J. R., Janssens-Maenhout, G. and Parker, R. J.: Global distribution of methane emissions, emission trends, and OH concentrations and trends inferred from an inversion of GOSAT satellite data for 2010-2015, Atmos. Chem. Phys., 19(11), 7859–7881, doi:10.5194/acp-19-7859-2019, 2019.

Zhang, Y., J. Jacob, D., Lu, X., D. Maasakkers, J., R. Scarpelli, T., Sheng, J. X., Shen, L., Qu, Z., P. Sulprizio, M., Chang, J., Anthony Bloom, A., Ma, S., Worden, J., J. Parker, R. and Boesch, H.: Attribution of the accelerating increase in atmospheric methane during 2010-2018 by inverse analysis of GOSAT observations, Atmos. Chem. Phys., 21(5), 3643–3666, doi:10.5194/acp-21-3643-2021, 2021.

---

## Author Response (AR2)

We thank again the reviewers for their comments. Our pointwise responses (AR) to reviewers' comments (RC: in *Italics*) and the respective changes in the manuscript (blue text) are as follows:

**Reviewer #1**

RC*: The authors have addressed and answered to the reviewers' comments carefully, and revised the manuscript accordingly. The manuscript is improved well, and I suggest it to be published after revising a few minor points below.*

*P1 L20: "as it accounts for" -> ", accounting for"*
*P4 L120: "be chemically removed" -> "be reduced (or become negligible?) due to atmospheric chemistry"*
*P4 L126: "a unit initial mole fraction" -> "an uniform initial mole fraction"?*
*P7 L208: "methane. sufficient" -> "methane, sufficient"?*
*P7 L218: "RMSD (root mean square difference)" -> "root mean square difference (RMSD)"*
*P7 L222: "prior observation RMSD"*
*Do you mean "prior (observation-prior?) RMSD", "observation RMSD" or something else?*
*P8 L238: "and over TRANSCOM regions" -> "and TRANSCOM regions."*
*P8 L243: "for a prior of" -> "with the prior emission of"*
*P8 L234: Please add +/- sign before "0.5 Tg" to be consistent with other adjustment numbers.*
*P8 L247: "The South American temperate region has the largest difference between the serial and PP emission estimates of*
*2 Tg yr-1"In the previous sentence, you mention that the differences in Eurasian temperate is also 2 Tg. Please modify the sentence to make it clear what you meant to say.*
*P9 L280-281: Please merge with the previous paragraph. One-sentence paragraph is not appropriate.*
*Figure 1 caption. Please add a note that the diagram is an example when splitting the inversion into three blocks or modify the diagram to be more general with k blocks (e.g. H1, H2, H3 -> H1, H2,..Hk).*
*Figure 5 caption: "emission estimates of" -> "emission estimates from"*
*Figure 5 caption: "(see Figure 1)"*
*Did you really meant to refer to Figure 1 or regional definition illustrated in Figure 2?*

AC: We have corrected the manuscript following the suggestions of the reviewer.

**Reviewer #2**

*I have found the manuscript has greatly improved from the previous one.*
*Therefore, I recommend this for publication after minor corrections suggested below.*

*Minor comments:*
*L34: "CTM's" => "CTMs"*

*L54-55: "The method has the advantage… non-linear inverse problems."*
*Please note that even a variational method assumes linearity in its algorithm. Therefore, I do not agree with this statement.*

AC: The variational method can be applied to a weakly non-linear inverse problem when combined with an appropriate steepest-decent numerical minimizer. For example, Pandey et al (2015, 2016), Krol et al. (2013) and Naus et al. (2021) have performed non-linear inversions using the variational method. They use the M1QN3 minimizer, based on a quasi-Newtonian algorithm, which is more suitable for non-linear inversions (Gilbert and Lemaréchal, 1989).

We have modified the sentence and listed some of these studies:

"The variational approach can be applied to weakly non-linear inverse problems using a suitable steepest-decent numerical minimizer (Naus et al., 2021; Pandey et al., 2016)"

*L116: "$|x|$ denotes the global sum" I think this is very confusing because "|.|" is usually used for representing absolute*
*values.*

AC: We have removed $|x|$ from the equations. We now denote the summation matrix **E**.

"Here **E** denotes a summation matrix used to compute global sum of the elements of $x_l$."

*L145: "The correct adjoint implementation" => "The correctness of the adjoint implementation"*
*Eq. (9): How do you verify the equality? Should the equation be satisfied within round-off errors?*

AC: The adjoint test checks if the equality (Equation 9) is satisfied to an accuracy near the computing precision. We have added the information to the revised manuscript.

"The test checks if the equality

$$\langle M(\boldsymbol{a}), \boldsymbol{b} \rangle = \langle \boldsymbol{a}, M^*(\boldsymbol{b}) \rangle \ \text{.......} \quad (9),$$

is satisfied to an accuracy near the computing precision"

*L157: "This can be avoided by taking a …" Please elaborate how "another inversion" is prepared in advance".*

AC: We have added the following text to the revised manuscript:

"This can be avoided by taking a realistic $c_0$ from the posterior mole fractions simulations of another inversion covering the period before the PP inversion. If such an inversion is not available, $c_0$ can be computed by performing an inversion for the 1-year period preceding the PP inversion."

*L179-182: I have a concern about transport biases induced by a different resolution.*
*If you changed the model resolution, some bias (persistent difference from the original resolution) could arise (e.g.,*
*exchange rate between the upper-troposphere and the lower-stratosphere).*

AC: We have removed this text from the revised manuscript to address the issue raised by the reviewer.

*L208: "methane. sufficient for our test inversion" typo?*

AC: We have corrected the typo.
*3.2 Wall time: Could you specify the CPU you used here?*

AC: We used "12-core 2.6 GHz Intel Xeon E5-2690 v3".

*L281: "conventional" may be better than "traditional"*
*Also, I'm wondering if "11-year" is traditional…*

AC: We have changed the text to avoid confusion.

*L283-289: Please consider to make this paragraph to an independent subsection (i.e. 4.1 ).*

AC: Done.

*The authors present an implementation of a so-called physical parallelization (PP) for variational flux inversions: from a previously described PP aimed at carbon dioxide (CO2), they add developments to take into account the chemical reactivity of methane (CH4).*

*General comments*

*As already stated in the review of the first version of the paper, the developments described in this paper are particularly relevant since long-term methane inversions are now run by several teams and the issue of the trends in methane emissions by various types of sources is still under study. The revisions made after the first review have lead to a clearer introduction and description of the work. Technically, the notations are now clearer in the mathematical description. The writing has been improved but I think, although I am not an native English speaker, that some small mistakes remain.*

*The introduction to the paper is now clear and focuses on the main relevant points so that the reader understands why this implementation of PP is interesting for methane inversions at the global scale.*

*In Section 2 Physical parallelization for methane inversions, it is now clear which parts are the general or Chevallier (2013) developments and which are specific to this work and therefore, to methane.*

*In Section 3 PP Performance test, only the prior uncertainties are now used, which makes the message simpler.*

*Section 4 Discussion contains elements which have been asked for by the reviewers of the first version and makes clear the potential of the PP.*

*Specific comments*

*Section 1 Introduction*
*- p.2 l.36: "Inversions have been performed on multidecadal scales to assess the information content of long records of methane mole fractions." This sentence is a bit strange to me because running an inversion means that it is assumed that the assimilated data does actually contain information on the fluxes (and other variables). It would be good to be able to assess the information content prior to running inversions, which is sometimes approximated by using sensitivity studies.*

AC: We have modified the sentence:

"A few studies have performed inversions on multidecadal scales to constrain emissions using the long measurement record of methane mole fractions."

*Section 2 Physical parallelization for methane inversions*
*- p.3 l.81: the choice of x^a for the prior is not very good because in the notations used for analytical inversions, ^a denotes the posterior. Usually, the prior state vector is noted x^b.*

AC: We have replaced x^a with x^b.

*- p.4 l.108: "the emission differences between the prior and the iteration in the period preceding the block": it is not very clear what is preceding what: is it the same iteration but for the preceding period of time? the same period of time but the previous iteration? the previous iteration in the preceding period of time?*

AC: We have modified the text:

"Here the scalar $n_k^i$ accounts for the global mean mole fraction changes due to emission differences $(\boldsymbol{x}^i - \boldsymbol{x}^b)$ during the inversion period that precedes the block $k$."

*- p.4 l.114 Eq. 5: the [ and ] do not seem to be necessary, they give the idea that f is a function of x - x^a.*

AC: Following a suggestion of the 2^nd reviewer, we have removed the |x | notation. We need to use "[ ]" to denote
precedence of operations in the updated equation. We denote functions using "()", for example, in Equation 4.

*- p.4 l.123-124: "implementing an atmospheric sink operator S": here, either give more details on what is in this operator or put a reference to the section which contains the information.*

AC: Done

*- p.4 l.122: Fig. 1 is not totally clear and consistent with the text here. The colours red and green in Fig. 1 are not explained in its legend so that a first idea of the reader is that what is in red comes from Chevallier (2013) and what is in green is linked to this work.*

*But the last sentence of the legend states that it is the blue boxes which indicate which part comes from which work.*
*Nevertheless, the "corrections calculation" is from Chevallier (2013) according to Fig. 1 but from this work Please make the text and figure very clear on which parts come from whose work.*

AC: The "corrections calculation" step was implemented by Chevallier (2013) as "global mass increment" (see Equation 4).

We have modified the text to clear other confusions:

"In an iteration, the block mole fractions for the iteration $\boldsymbol{m}_k^i$ is computed using the block CTM operator $H_k$, the iteration
emissions for the block $\boldsymbol{x}_k^i$, the initial mole fraction for this block $\boldsymbol{c}_k^b$, and a mole fraction correction $n_k^i$"

We have added the following to the legend of Figure 1:
"The steps shown with red boxes use CTM runs and take long wall time. The steps shown in green are without CTM runs and require negligible wall time

*- p.5 l.131: "Sl,k accounts for the impact of atmospheric sinks": the reader still does not know at this point what is in S and the link to Sl,k. A paragraph dedicated to the sink, which is the main feature of the PP elaborated here is required at some point.*

AC: We clarified and added more information on the $S$:

"$n_k^i = h_k \sum_{l=1}^{k-1} s_{k,l} \; f \; E \left[ x_l^i - x_l^b \right]$ ...........(6).

Here the scalar $s_{k,l}$ accounts for the impact of atmospheric sinks on the global uniform mole fraction change during the time period between the blocks k and l. $s_{k,l}$ is generated using a sink operator S. We describe a formulation of S in the next section."

"We parameterize the sink operator $S$, which computes the sink scaling factor $s_{k,l}$ (Equation 6), with an e-folding decay function and a constant atmospheric lifetime of methane ($\tau$) of 9 years.
$s_{k,l} = S(k,l) = e^{-|t_l - t_k|/\tau}$ ....... (10)

Here $t_l$ and $t_k$ are the start times of the blocks $l$ and $k$, respectively. We found this simple parameterization with a constant lifetime is sufficient for our test inversion. "

*- p.5 l.143 Eq 8: the notations are not clear to me in this equation. The adjoint of S is denoted by \*, which indicates that it is not assumed to be linear. But the adjoint of h is indicated by T, which indicates that it is linear. This is not totally consistent*
*with H begin the CTM operator, with an adjoint denoted by \* in Eq. 3 and p.5 l.138 above. Could you provide more details on the derivation of the equations and make the notations totally consistent?*

AC: For $S$ operator, see our response to the previous comment. We have clarified $h_l^T$ in the revised manuscript

"$g_k^i = f \sum_{l=k+1}^{r} s_{k,l} \; h_l^T \; \delta m_l^i$        (8),

Here $h_l^T \; \delta m_l^i$ is the matrix dot product of the two vectors, both of which have the same size"

*- p.6 l.161 seq.: the summary of the practical steps of the PP is a good idea but there is nothing about the sink. Please add the necessary steps since it is the main feature of the PP described here.*

AC: We have added the additional step to the summary.

"Prepare a sink operator $S$ which accounts for the impact of atmospheric sinks on methane mole fractions during a period."
*Section 3 PP Performance test*
*- p.7 l.215: the title of the subsection is "Emission estimation errors" but the first two paragraphs are about the match in the concentration space. Maybe it would be clearer to make a subsection dedicated to the concentrations separated from the subsection on the emissions.*

AC: Done.

*- p.8 l.228: "For both inversions, a good model fit to the observations" Shouldn't "a good fit" be defined, if possible with reference to R?*

AC: We have now defined a good fit as "90 % reduction in mean of observation-model mismatch".

- *p.8 l.237: "should be in good agreement" Same remark as above: how is a "good" agreement defined?*

AC: We have quantitively described the PP errors in the Section 3.1.2. We have modified the sentence to clarify:

"The physically parallelized CTM used in the PP inversion has lost some of the consistency of the full CTM used in the serial inversion and the PP emission errors will depend on the impact of this CTM simplification."

- *p.8 l.237-238: "from the inversions integrated over the globe and over TRANSCOM regions". According to the description of B (p.7 l.195 seq.), the inversions are performed at the pixel's resolution. It is a bit strange therefore to assess the performances of the PP using (very) large regions. I guess a lot of inversions can be in "good" agreement over the whole globe since the constraint on the mean total emissions of methane is strong (as stated l.240-241 "The global methane*
*emissions are in general well constrained by the NOAA observations in a serial inversion"). The same applies for large regions such as TRANSCOM's: for example, an inversion with homogeneous emissions inside a region and an inversion with a dipole of negative/positive increments can give the same total over a large region. I understand that it is not simple to compare two inversions at the grid cell's resolution but showing maps would seem to me to be more relevant than the average global total.*

AC: Inversion studies of methane (and $CO_2$) assess emissions at regional scales because of the reasons that the reviewer has stated. Also, we use NOAA background observation in our test inversions, which does not provide a good constraint on the grid scale emissions. Although, we believe that the assessment of the emissions as global and TRANSCOM regions total is more relevant than grid cell resolution, we have added a figure to the manuscript showing the emission differences at grid scale following the suggestion of the reviewer:

[Figure]

[Figure]

**"Figure 8 Figure 8** Emission differences averaged over 1999-2010. Panel (a) shows differences between serial and prior. Panel (b) shows differences between serial and PP"

AC: We have added the following text to the Section 3.2.1:

"Figure 8 shows the spatial distribution of the emission differences at grid scale. The mean ($\pm$ 1$\sigma$ spread) of the differences between the serial inversion and prior is $-8 \times 10^{-3}$ ($\pm$ 0.5 ) Tg gridbox$^{-1}$ yr$^{-1}$, and it is $9 \times 10^{-5}$ ($\pm$ 0.04 ) Tg gridbox$^{-1}$ yr$^{-1}$ for serial and PP inversions. Emission differences between the PP and serial inversions are visible over India and South American temperate. These differences are likely due to the lack of observational constraint in these regions (see Figure 2)."

*- p.8 l.241-242: "the additional error introduced by the PP method does not seem to have a significant impact on the global emissions': please define "significant".*

AC: We have made the sentence more quantitative:

"The global methane emissions are in general well constrained by the NOAA observations in a serial inversion, and the
additional error introduced by the PP method only causes a 1 % error relative to the serial-prior emission mismatch."

*- p.8 l.244: "is well within the prior emission uncertainty". I don't understand the argument here. The difference between the two inversions should be such that they are inside each other's (posterior) uncertainty; the prior uncertainty is what the inversion aims at reducing so it does not seem to be a good measure for a "small" difference.*

AC: We do not have a good estimate of the posterior uncertainties. Therefore, we evaluate the PP-serial emission differences relative to prior uncertainties. We have added the following text to clarify:

"We do not have a good estimate of the posterior uncertainties because a large number of variational inversion iterations are needed for the second derivative of the cost function to converge. Therefore, we evaluate PP performance by comparing the PP-serial emission differences against the emission adjustments performed by the serial inversion (serial-prior differences)

and prior emission uncertainties."

*- p.8 l.249: "estimates for the TRANSCOM regions deviate within < 5 % from the serial emissions": how can you convince the reader that 5% is a small difference?*

AC: We have modified this sentence in the revised manuscript as follows:

"In summary, mean PP emission estimates for the TRANSCOM regions deviate within < 5 % from the prior emissions, while the serial-prior differences are up to 50 % of the prior emissions."

*- p.9 l.260: "the PP inversion can effectively reproduce results of the serial inversion" but only at a coarser resolution than the actual control vector, according to what is shown previously. This sentence is too optimistic as such.*

AC: We have modified the sentence to address the concern of the reviewer:

"…shows that the PP inversion can effectively reproduce results of the serial inversion at regional scales"

*- p.9 l.265-266: "a forward or adjoint TM5 CTM run of one year took about 15 minutes": it is a bit strange that the adjoint and forward codes take the same time to run - usually, the adjoint takes more time as it requires to recompute (or at least*

*read) parts of the forward in sub-time-steps.*

AC: The TM5 run used in this study is strictly linear. Therefore, the adjoint CTM run does not read or recompute parts of the forward CTM in sub-time-steps.

*- p.9 l.270 seq.: the times for each step are indicated but nothing is said on the sink (see also comments on Section 2). Please*

*add the information, even if it is only to state that it takes a very small computing time.*

AC: We have added that the sink operater takes negligible time.

*Section 4 Discussion*
*- p.10 l.307-309: "the need for a serial sequence of inversions to provide a time series of initial mole fractions imposes a*

*limitation to the model resolution that can be used." This sentence is not very clear to me. Do you mean that the model's resolution must not be too fine so that the serial sequence of inversions does not take too long?*

AC: We have modified the sentence:

"These numbers depend of course on the parallel efficiency of the model and the computing server. The need for a coarse resolution serial sequence of inversions to provide initial mole fractions fields limits the inversion period for which this method can be used."

*- p.10 l.310: "the wall time performance of the CAMS reanalysis inversions will improve in future". Taking into account the estimated gains in computing time given in Section 3, is it possible to give an order of magnitude of this expected*
*improvement?*

AC: We expect the wall time performance of a 30-year CAMS reanalysis inversion to improve by 5 to 10 times relative to the current CAMS inversion setup (which is not a serial inversion) if the PP implementation is similar to that used in this study: splitting the inversion period into annual block with 9-month overlaps. The actual improvement will depend on the number of sensitivity tests and checks performed and will be known after those CAMS inversion are performed.

*- p.11 l.323: "optimizing emissions from large TRANSCOM regions". I understood from the description of the B matrix that the emissions were optimized at the resolution of the model's pixel (see also my comments on p.8 l.237-238). Here, it looks like the control vector contains the TRANSCOM regions. Please clarify.*

AC: The inversion is performed at model pixel resolution. The emission estimates from the inversion are evaluated at
TRANSCOM region scales. We have modified the sentence to address the confusion:

"We used a 9-month overlap in our test inversion setup. It was sufficient estimate the total emissions from TRANSCOM regions using the surface observations"

*- p.11 l.334-336: "Furthermore, the performance gained by performing the inversions at higher resolution because of the*
*improved computational performance will likely outweigh the accuracy loss due to the assumptions made in the PP method." This does not seem so likely to me. Do you have any references or examples of such a positive case?*

AC: We do not have a reference for this. We assume that performing the inversion at a higher resolution will reduce the following errors:

1. Aggregation error: error in the distribution of emission at for example 1 x 1 degree grid that is not corrected by
adjustment of the emissions state vector at 6 x 4 degree grid scale.

2. Model representation error: spatial and temporal smoothing intrinsic to the model resolution which prevents it from resolving finer-scale variability in the observations.

If PP method is implemented properly, with sufficient overlap and a sink operator S, the errors in PP should remain smaller than these errors.

*- p.11-12 l.348-349: "These studies assume a quasi-linearity for the inversion as changes to the methane mole fractions are expected to remain small compared to the mean." This is not a definition of (quasi-)linearity. Please elaborate.*

AC: We have added the following text to elaborate on this:

*"The simultaneous optimization of OH with methane emissions introduces a non-linearity in the inversion because methane loss rate depends on the product of methane and OH mole fractions. However, the changes to the methane mole fractions are expected to remain small during the inversions. Hence, the non-linear effect is small and a quasi-linearity is assumed to solve the inversion analytically using the computation of the full Jacobian matrix of the CTM. Under a quasi-linearity assumption, OH can be optimized in a PP methane inversion by introducing annual OH scaling factors in the state vector and the methane lifetimes in the sink operator can be scaled in each iteration to reflect the corresponding OH adjustments."*

*Section 5 Conclusions*
*- p.12 l.355: "An atmospheric inversion with a very large state vector is needed" This is not exactly true: many inversions are run with large regions and are able to use the available information on the tendencies, the North-South gradient, etc.*

AC: We have modified the sentence:

"An atmospheric inversion with a very large state vector is needed optimize emissions using such long measurement records at a grid scale"

*Figure 5*
*What is the link between the "2-sigma uncertainties of the prior emissions" and the description of B (p.7 l.195 seq.) i.e. 50% per grid cell per month plus the covariances (spatial and temporal correlations)?*

AC: Yes, the $\pm 2\sigma$ prior uncertainties of the large regions shown in Figure 5 account for the spatial and temporal correlations (off-diagonal element of **B**) between the cells. We have modified text to clarify how **B** is constructed

"The prior covariance matrix **B** is constructed as follows. The diagonal elements of **B** are constructed assuming $\pm 1\sigma$
uncertainties of 50 % of the emissions per grid cell per month. The off-diagonal elements are constructed by assuming the emissions to be correlated temporally using an exponential correlation function with an e-folding time scale of 3 months, and spatially with a Gaussian correlation function using a length scale of 500 km (Houweling et al., 2014)."

*Technical corrections*
*- in the introduction (and maybe other places also): "CTM's" (saxon genitive, I guess) must be changed to "CTMs" (plural of CTM)*
*- p.2 l.63: "emissions adjustments" -> emission adjustments?*
*- p.4 l.122: "scaler" -> scalar?*
*- p.4 l.124: "we use a CTM block sensitivity vector $h\_k$ distribute global emission changes more precisely" -> we use a*
*CTM block sensitivity vector $h\_k$ TO distribute THE global emission changes more precisely?*
*- p.7 l.200: "The emissions in 2008 applied to every year in the inversion period" -> The emissions of the year 2008 are used for every year of the prior?*
*- p.7 l.208: ". sufficient for our test inversion (Section 3)" -> does this part of sentence goes with the previous one? If so, change "." to ",". Also, why is there a reference to Section 3 inside Section 3? Please check.*
*- p.7 l.224: "as good as" -> as well as?*

AC: We have implemented the technical corrections suggested by the reviewer.

**References**

Gilbert, J. C. and Lemaréchal, C.: Some numerical experiments with variable-storage quasi-Newton algorithms, Math. Program., 45, 407–435, 1989.

Krol, M. C., Hooghiemstra, P. B., van Leeuwen, T. T., van der Werf, G. R., Novelli, P. C., Deeter, M. N., Aben, I., & Röckmann, T. (2013). Correction to "Interannual variability of carbon monoxide emission estimates over South America from 2006 to 2010." *Journal of Geophysical Research: Atmospheres*, *118*(10), 5061–5064. https://doi.org/10.1002/jgrd.50389

Naus, S., Montzka, S. A., Patra, P. K., & Krol, M. C. (2021). A three-dimensional-model inversion of methyl chloroform to constrain the atmospheric oxidative capacity. *Atmospheric Chemistry and Physics*, *21*(6), 4809–4824. https://doi.org/10.5194/acp-21-4809-2021

Pandey, S., Houweling, S., Krol, M., Aben, I., & Röckmann, T. (2015). On the use of satellite-derived $CH_4$ : $CO_2$ columns in a joint inversion of $CH_4$ and $CO_2$ fluxes. *Atmospheric Chemistry and Physics*, *15*, 8615–8629. https://doi.org/10.5194/acp-15-8615-2015

Pandey, S., Houweling, S., Krol, M., Aben, I., Chevallier, F., Dlugokencky, E. J., Gatti, L. V., Gloor, E., Miller, J. B., Detmers, R., Machida, T., & Röckmann, T. (2016). Inverse modeling of GOSAT-retrieved ratios of total column $CH_4$ and $CO_2$ for 2009 and 2010. *Atmospheric Chemistry and Physics*, *16*(8), 5043–5062. https://doi.org/10.5194/acp-16-5043-2016